# DNA-dependent protein kinase catalytic subunit (DNA-PKcs) drives chronic kidney disease progression in male mice

Yunwen Yang [1,2,3], Suwen Liu [4], Peipei Wang [1,2,3], Jing Ouyang [1,2,3], Ning Zhou [1,2,3], Yue Zhang [1,2,3] ✉, Songming Huang [1,2,3] ✉, Zhanjun Jia [1,2,3] ✉ & Aihua Zhang [1,2,3] ✉

Kidney injury initiates epithelial dedifferentiation and myofibroblast activation during the progression of chronic kidney disease. Herein, we find that the expression of DNA-PKcs is significantly increased in the kidney tissues of both chronic kidney disease patients and male mice induced by unilateral ureteral obstruction and unilateral ischemia-reperfusion injury. In vivo, knockout of DNA-PKcs or treatment with its specific inhibitor NU7441 hampers the development of chronic kidney disease in male mice. In vitro, DNA-PKcs deficiency preserves epithelial cell phenotype and inhibits fibroblast activation induced by transforming growth factor-beta 1. Additionally, our results show that TAF7, as a possible substrate of DNA-PKcs, enhances mTORC1 activation by upregulating RAPTOR expression, which subsequently promotes metabolic reprogramming in injured epithelial cells and myofibroblasts. Taken together, DNA-PKcs can be inhibited to correct metabolic reprogramming via the TAF7/ mTORC1 signaling in chronic kidney disease, and serve as a potential target for treating chronic kidney disease.

Chronic kidney disease (CKD) inflicts on 10% of the adults worldwide[1]. CKD poses a huge burden on global public health, as it has a high likeliness to progress into end-stage renal disease (ESRD), which requires dialysis or kidney transplantation[1,2]. Renal interstitial fibrosis is one of the major pathological characteristics of CKD[2–4]. Many of the pathophysiological features of kidney fibrosis are shared by other fibrotic diseases, such as liver cirrhosis and cardiomyopathies[5]. There are no effective treatments for CKD, highlighting an urgent need for a better understanding of the pathological mechanisms underlying CKD.

In injured kidneys, interstitial fibrosis involves the abnormal expression of profibrotic factors, such as transforming growth factor-beta 1 (TGF-β1), epithelial dedifferentiation and myofibroblast activation[2]. As a key mediator of interstitial fibrosis, TGF-β1 not only activates the expression of fibrotic genes, including α-smooth muscle actin (α-SMA), fibronectin (FN) and collagens, but also promotes Warburg effect-like metabolic reprogramming of kidney cells[6,7]. Recently, emerging evidence has demonstrated the relationship between metabolic dysregulation and interstitial fibrosis, revealing that metabolic reprogramming occurs in kidney cells (renal tubular epithelial cells[8,9] and myofibroblasts[10]) during kidney injury and contributes to the development of CKD. Metabolic reprogramming in kidney cells causes a drastic reduction in fatty acid oxidation (FAO) and a metabolic shift to glycolysis, contributing to immune cell infiltration and interstitial fibrosis[7,11,12]. Furthermore, clinical data have revealed that metabolism and inflammation pathways are dysregulated in CKD patients[7]. Restoring FAO or inhibiting glycolysis through genetic or pharmacological methods attenuates fibrosis in various animal models of renal fibrosis[11,13–15].

[1]Department of Nephrology, Children's Hospital of Nanjing Medical University, Guangzhou Road #72, Nanjing 210008, China. [2]Nanjing Key Laboratory of Pediatrics, Children's Hospital of Nanjing Medical University, Nanjing 210008, China. [3]Jiangsu Key Laboratory of Pediatrics, Nanjing Medical University, Nanjing 210029, China. [4]Department of Pediatrics, Shandong Provincial Hospital Affiliated to Shandong First Medical University, Jinan 250021, China. ✉e-mail: zyflora2006@hotmail.com; smhuang@njmu.edu.cn; jiazj72@hotmail.com; zhaihua@njmu.edu.cn

In the molecular mechanisms of metabolic reprogramming, mammalian target of rapamycin (mTOR), which is a cytoplasmic serine/threonine protein kinase, acts to modulate cell metabolism and maintain metabolic homeostasis[16,17]. Two distinct functional complexes of mTOR have been defined according to its binding proteins. MTOR complex 1 (mTORC1) interacts with its specific regulator RAPTOR and is sensitive to rapamycin[18,19]. mTOR complex 2 (mTORC2) is regulated by RICTOR[20]. Chronic mTORC1 activation promotes metabolic reprogramming in various pathological conditions, including renal interstitial fibrosis[9,21–23], but the mechanisms that regulate this activation, especially in interstitial fibrosis, remain largely unknown.

DNA-dependent protein kinase (DNA-PK), a trimeric complex composed of a catalytic subunit (DNA-PKcs) and a Ku70/80 heterodimer, is activated by DNA double-stranded breaks (DSBs)[24,25] or ROS[26]. A well-known function of DNA-PK is mediating nonhomologous end joining (NHEJ), which joins programmed DSBs created during V(D)J recombination and plays a key role in the recombination of lymphocytes[25]. As a result, mutations in DNA-PKcs arrest the development of T and B lymphocytes, due to the deficiency of V (D) J recombination[27,28]. However, the growth retardation or high frequency of T cell lymphoma development did not appear in DNA-PKcs knockout mice[28]. Although it is a DNA DSB sensor, DNA-PKcs has some unusual properties, such as accumulating in cells to a level far above what is likely needed for NHEJ[29]. Moreover, DNA-PKcs is located both in the nucleus and the cytoplasm[30]. In agreement with its unusual properties, recent evidence indicates that DNA-PKcs has additional functions other than NHEJ[31]. For example, DNA-PKcs-mediated phosphorylation of the transcription factor USF-1 promotes fatty acid synthesis induced by insulin[32], and DNA-PKcs promotes metabolic decline during aging[33]. Research on the non-NHEJ functions of DNA-PKcs, such as metabolism and aging, has just budded, and a number of questions remain unanswered[31]. Additionally, as one of the phosphoinositide 3-kinase (PI3K)-related kinases, DNA-PK had been found to regulate mTOR activation[34].

In this work, our results show that DNA-PKcs mediates the activation of RAPTOR/mTORC1 signaling through phosphorylation of TATA-box binding protein associated factor 7 (TAF7) in CKD. Inhibition of DNA-PKcs corrects metabolic reprogramming in injured epithelial cells and myofibroblasts in CKD. DNA-PKcs may serve as a new potential target for treating CKD.

## Results

### DNA-PKcs is increased in the kidneys of CKD patients and in mice with kidney fibrosis

The activity of DNA-PKcs was analyzed by immunohistochemical staining of S2056 autophosphorylation in DNA-PKcs (p-DNA-PKcs) in healthy and CKD human kidney tissues. The nontumor portions of nephrectomy samples were used as "healthy" kidney tissues, which were collected from patients with renal cell carcinoma. The results showed that p-DNA-PKcs was almost undetectable in these normal kidney tissues, but the protein levels of p-DNA-PKcs were markedly increased in CKD human kidney tissues (Fig. 1a). To determine the relevance of DNA-PKcs activity and renal fibrosis in vivo, expression of fibronectin (FN), a crucial gene related to kidney fibrosis, was analyzed by immunohistochemical staining, and Sirius red staining was also performed to analyze the degree of interstitial fibrosis in CKD patients (Fig. 1a). Our results revealed a positive association between the protein levels of p-DNA-PKcs and the degree of interstitial fibrosis ($P = 0.005$, Fig. 1b, c). Additionally, expression levels of DNA-PKcs were also analyzed using previously published human CKD microarray datasets (Nephroseq). We found that mRNA levels of DNA-PKcs were significantly increased by more than 1.6-fold in renal biopsy tissues of 48 CKD patients compared to controls (Supplementary Fig. 1a), exhibiting a significant positive correlation with the degree of kidney fibrosis in patients (Supplementary Fig. 1b). Furthermore, the results of

immunofluorescence co-staining of p-DNA-PKcs with Lotus tetragonolobus lectin (LTL) (a renal tubule marker) or α-SMA (a marker of myofibroblasts) showed that DNA-PKcs was upregulated in both tubular epithelial cells and myofibroblasts of the kidney tissues from CKD patients (Fig. 1d, e). Analysis of an available online renal single-cell RNA sequencing database (http://humphreyslab.com/SingleCell/)[35] revealed that DNA-PKcs mRNA was widely expressed in kidney cells and upregulated in human diabetic kidneys compared to control kidneys (Supplementary Fig. 1c).

To further identify the association between DNA-PKcs activity and kidney fibrosis development, we next examined the expression and activity of DNA-PKcs in two mouse models of kidney fibrosis, unilateral ureteral obstruction (UUO) and unilateral ischemia-reperfusion (UIR). Immunohistochemical staining of p-DNA-PKcs indicated that p-DNA-PKcs was markedly induced on day 3 compared to sham controls in the UUO model, and protein levels of p-DNA-PKcs were further increased as kidney fibrosis developed from day 3 to day 14 (Fig. 1f). Western blot and qRT-PCR results showed that the total protein and mRNA levels of DNA-PKcs were also significantly upregulated in kidney tissues of UUO mice compared to sham controls (Fig. 1g and Supplementary Fig. 1e). Similar results were observed in UIR mice (Fig. 1h and Supplementary Fig. 1d, f). The potential link between DNA-PKcs and CKD prompted us to further investigate the role and mechanistic implications of DNA-PKcs in CKD.

### DNA-PKcs-mediated progression of CKD in mice

Next, we created DNA-PKcs knockout mice in which the DNA-PKcs-encoded gene *Prkdc* was knocked out using CRISPR/cas9 as described in the Methods (Supplementary Fig. 2a). DNA-PKcs heterozygous (+/−) and homozygous (−/−) knockout and wild-type (WT) mice were distinguished by genotyping, as shown in Supplementary Fig. 2a. *Prkdc* mRNA (Supplementary Fig. 2c) and DNA-PKcs protein levels (Fig. 2e) were not detected in kidney tissues of DNA-PKcs−/− mice, indicating successful knockout of DNA-PKcs in mice. To evaluate the effect of the DNA-PKcs knockout strategy on renal damage and fibrosis, adult (6–8 weeks old) DNA-PKcs−/− and WT control male mice were subjected to UUO and euthanized 7 days after surgery. PAS staining revealed that the degree of tubular atrophy and dilatation induced by UUO was markedly ameliorated in DNA-PKcs−/− mice compared to WT controls (Fig. 2a). Renal fibrosis was also greatly ameliorated in DNA-PKcs−/− mice compared to WT mice, as observed, and quantified by Masson staining (Fig. 2b). Images of immune labeling for interstitial collagen I deposition and α-SMA showed that both classical profibrotic markers were significantly reduced in the UUO kidneys of DNA-PKcs−/− mice compared to WT controls (Fig. 2c, d). Protein levels of profibrotic markers, including FN and α-SMA, were also significantly reduced in the UUO kidneys of DNA-PKcs−/− mice, as analyzed by Western blot (Fig. 2e). QRT-PCR analysis showed that knocking out DNA-PKcs in mice markedly reduced expression levels of crucial kidney fibrosis genes, including *Col1a1, Col3a1, Fn1,* and *Acta2,* which were greatly upregulated in UUO kidneys. Consistent with reduced fibrosis, DNA-PKcs knockout also reduced mRNA levels of the kidney injury marker KIM-1 induced by UUO (Supplementary Fig. 2c). Moreover, inflammation is a major hallmark of kidney fibrosis and plays an important role in the pathogenesis of renal fibrosis. Thus, we analyzed inflammation-related markers, including the cytokines *Il1β, Il6, Tnfα,* and *Mcp1,* by qRT-PCR. In the UUO model, we found that upregulation of inflammatory cytokines induced by UUO was markedly reduced by DNA-PKcs knockout (Supplementary Fig. 2c). As both clinical studies and animal models exhibit a strong correlation between macrophages and the extent of renal fibrosis, infiltrated macrophages in UUO kidney tissues were assessed by immunohistochemical staining for the macrophage marker F4/80. The images revealed many infiltrated macrophages in obstructed kidneys of WT mice 7 days after UUO, which were markedly decreased by DNA-PKcs knockout (Fig. 2f). Interesting, DNA-PKcs

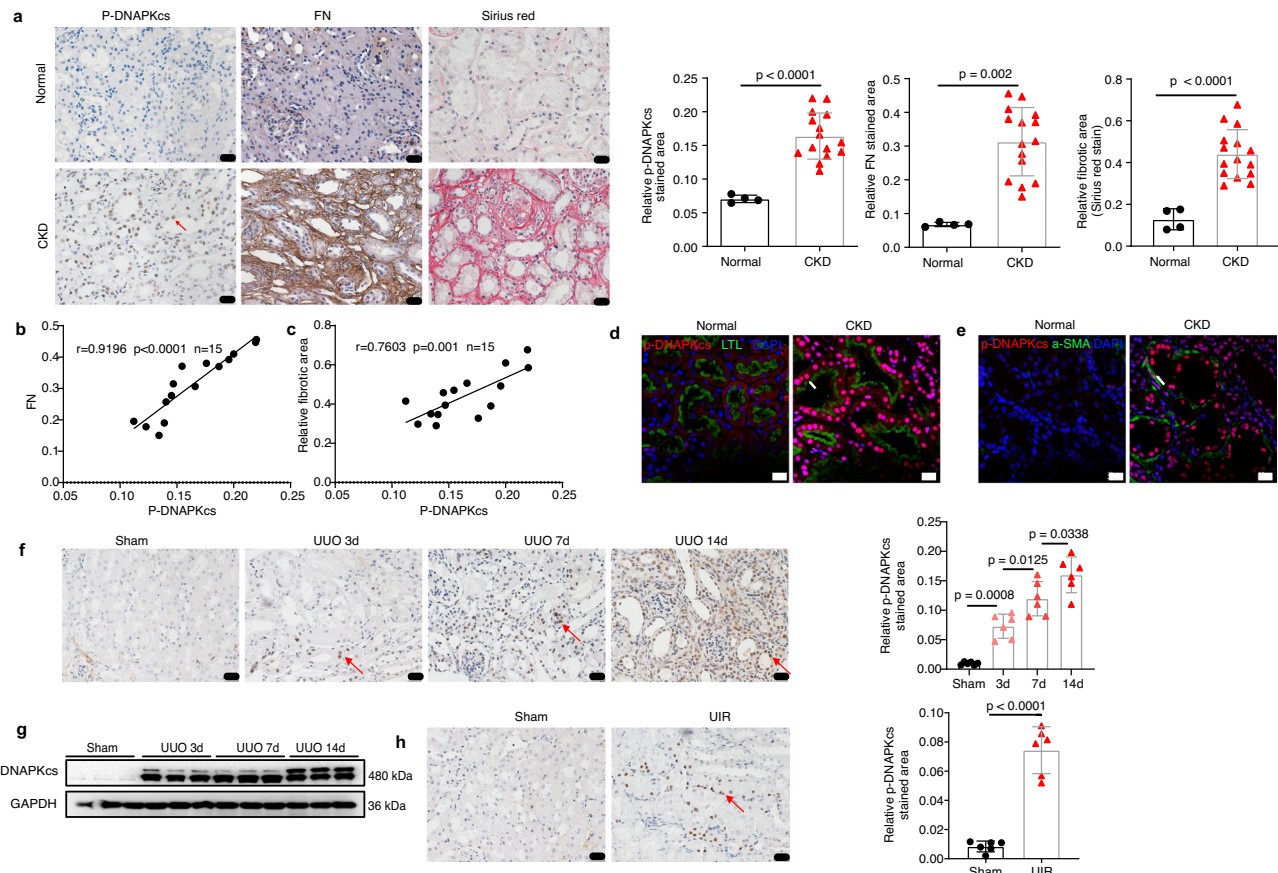

**Fig. 1 | DNA-PKcs is increased in the kidneys of CKD patients and mice with kidney fibrosis. a** Immunohistochemical staining of p-DNA-PKcs (indicated by red arrow), FN and Sirius red staining in kidney tissues of normal ($n = 4$) and CKD ($n = 15$) patients, scale bars: 20 μm. Bars represent quantification results and data are shown as the mean ± SD. Two-tailed unpaired t-test was used to determine the $p$-values. **b** Pearson's r correlation analysis between FN and p-DNA-PKcs protein levels in CKD patients ($n = 15$) with 95% confidence interval from 0.7701 to 0.9733. **c** Pearson's $r$ correlation analysis between collagens deposition (Sirius red) and p-DNA-PKcs protein levels in CKD patients ($n = 15$) with 95% confidence interval from 0.4062 to 0.9159. **d** Immunofluorescence co-staining of p-DNA-PKcs with a renal tubule marker, Lotus tetragonolobus lectin (LTL) in kidney tissues of normal ($n = 4$) and CKD ($n = 4$) human study participants, red: p-DNA-PKcs, green: LTL, bule: DAPI, scale bars: 20 μm. Tubule is indicated by arrow. **e** Immunofluorescence co-staining of p-DNA-PKcs with α-SMA in kidney tissues of normal ($n = 4$) and CKD ($n = 4$) human study participants, red: p-DNA-PKcs, green: a-SMA, bule: DAPI, scale bars: 20 μm. Myofibroblast is indicated by arrow. **f** Immunohistochemical staining of p-DNA-PKcs (indicated by red arrow) in kidney tissues of UUO mice, scale bars: 20 μm. Bars represent quantification results (mean ± SD, $n = 6$ mice of each group). One-way ANOVA with Tukey's multiple comparisons test was used to determine the $p$-values. **g** Western blot analysis of DNA-PKcs in kidney tissues of UUO mice ($n = 6$ mice of each group). **h** Immunohistochemical staining of p-DNA-PKcs (indicated by red arrow) in kidney tissues of UIR mice (day 21), scale bars: 20 μm. Bars represent quantification results (mean ± SD, $n = 6$ mice of each group). Two-tailed unpaired t-test was used to determine the $p$-values. Source data are provided as a Source Data file.

knockout didn't aggravate DSBs induced by UUO indicated by lower levels of γH2AX (Supplementary Fig. 2d).

To further assess the impact of renal tubular DNA-PKcs on renal fibrosis, we constructed proximal renal tubular epithelial cell with specific DNA-PKcs knockout in vivo by using CRISPR/cas9 knockin mice (Supplementary Fig. 2e, f). After subcapsular injection of adeno-associated virus (AAV) containing guide RNA (gRNA) for DNA-PKcs for three weeks, the proximal renal tubular cell-specific DNA-PKcs knockout mice were subjected to UUO and euthanized at 7 days after surgery. The immunofluorescence staining of DNA-PKcs showed an absence of DNA-PKcs in renal tubular cells (Fig. 2h) but it was still expression in α-SMA positive myofibroblasts (Supplementary Fig. 2g) of cas9 mice injected with AAV-gRNA, compared to those in cas9 mice injected with an AAV without gRNA for DNA-PKcs. Western blot analysis showed that the protein levels of DNA-PKcs and profibrotic markers, including FN and α-SMA, were also significantly reduced in the UUO kidneys of cas9 mice injected with AAV-sgRNA, compared to those in the control group (Fig. 2g). Similar to the UUO model, DNA-PKcs knockout mice were also protected from UIR-induced CKD (Supplementary Fig. 2h–m).

Together, these results suggest that deletion of DNA-PKcs attenuates the progression of CKD in mice.

To determine whether overexpression of DNA-PKcs in the kidney is sufficient to drive kidney fibrosis, human DNA-PKcs overexpression plasmids (Addgene: 83317) were delivered into mouse kidneys through a tail vein high-pressure injection method as described in several previous studies[36]. Thirty-six hours after DNA-PKcs plasmid injection, the mice were administered UUO surgery for 7 days. Collectively, our results indicate that ectopic expression of human DNA-PKcs is sufficient to drive the progression of renal fibrosis in vivo (Supplementary Fig. 3).

## Inhibition of DNA-PKcs activity attenuates the development of CKD in mice

These findings led us to consider the possibility that inhibitors of DNA-PKcs could be used to protect against renal fibrosis. To test our hypothesis, adult (6–8 weeks old) WT male mice were treated daily with the highly specific DNA-PKcs inhibitor NU7441 beginning 1 day after UUO surgery. NU7441-treated and control mice were euthanized 7 days after UUO surgery, DNA-PKcs activity was analyzed by

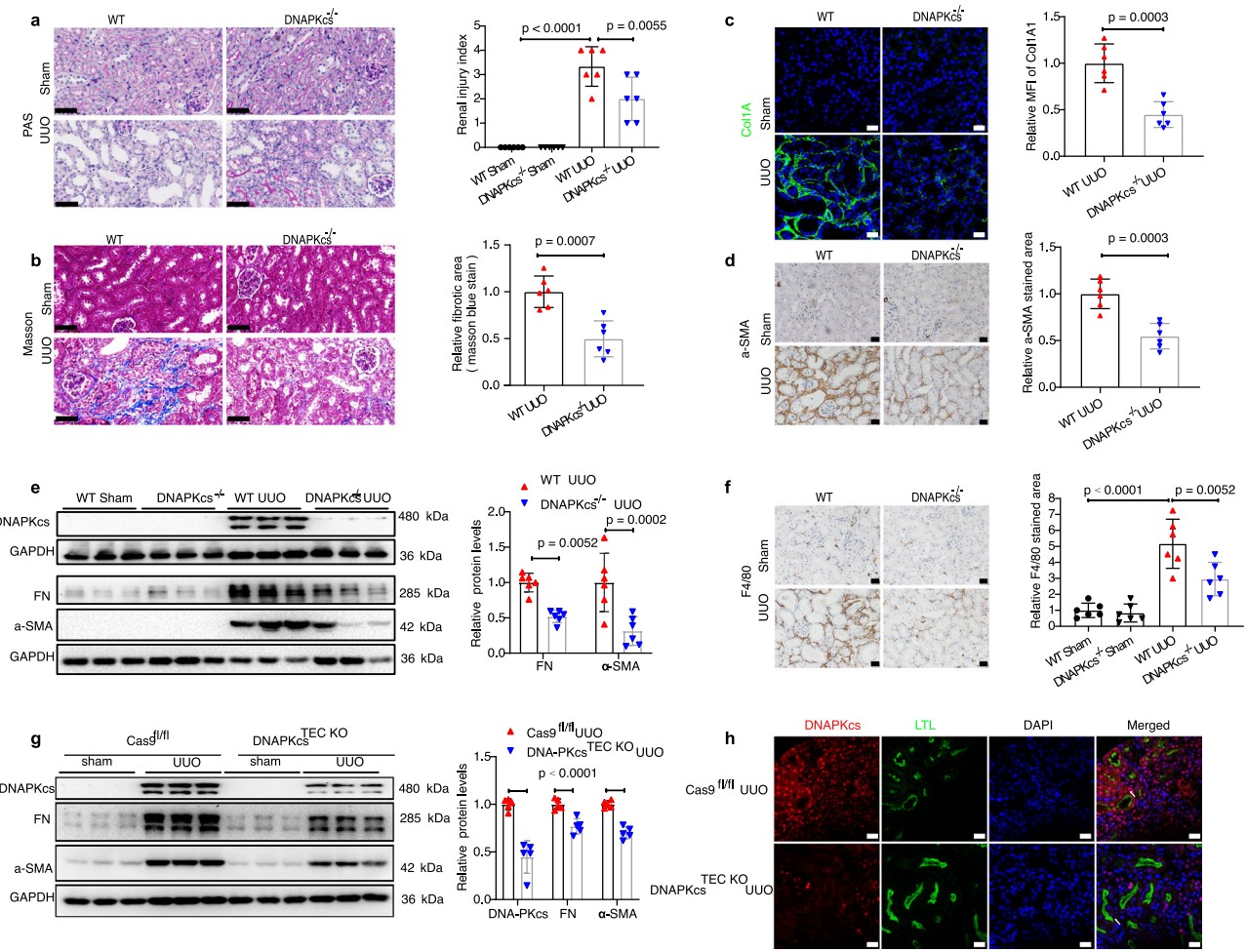

**Fig. 2 | Deletion of DNA-PKcs attenuates the progression of CKD in UUO mice.** **a** PAS staining of kidneys of WT and DNA-PKcs$^{-/-}$ mice subjected to UUO (day 7), scale bars: 50 μm. Bars represent quantification results (mean ± SD, $n = 6$ mice of each group). **b** Masson staining of kidneys of WT and DNA-PKcs$^{-/-}$ mice subjected to UUO (day 7), scale bars: 50 μm. **c** Immunofluorescence staining of COL1A1, scale bars: 20 μm, MFI mean fluorescence intensity. Bars represent quantification results (mean ± SD, $n = 6$ mice of each group). **d** Immunohistochemical staining of α-SMA in kidneys of WT and DNA-PKcs$^{-/-}$ mice subjected to UUO (bar: 20 μm). Bars represent quantification results (mean ± SD, $n = 6$ mice of each group). **e** Protein levels of DNA-PKcs, FN and α-SMA in kidneys of WT and DNA-PKcs$^{-/-}$ mice subjected to UUO. Bars represent quantification results (mean ± SD, $n = 6$ mice of each group). **f** Immunohistochemical staining of F4/80 in kidneys of WT and DNA-PKcs$^{-/-}$ mice subjected to UUO (bar: 20 μm). Bars represent quantification results (mean ± SD, $n = 6$ mice of each group). **g** Protein levels of DNA-PKcs, FN, and α-SMA in kidneys of tubular epithelial cells (TEC) specific DNA-PKcs$^{-/-}$ and control mice subjected to UUO (day 7). Bars represent quantification results (mean ± SD, $n = 5$ mice of each group). **h** Immunofluorescence staining of DNA-PKcs and LTL in kidneys of DNA-PKcs$^{TEC\ KO}$ and control mice subjected to UUO (day 7, $n = 5$ mice of each group), scale bar: 20 μm. Two-tailed unpaired $t$-test was used to determine the $p$-values for **b**–**d**. One-way ANOVA followed by Tukey's multiple comparisons test was used to determine the $p$-values for **a**, **f**. Two-way ANOVAs followed by Šídák's multiple comparisons test were used to determine the $p$-values for **e**, **g**. Source data are provided as a Source Data file.

immunohistochemical staining, and the results showed that NU7441 treatment markedly inhibited the activation of DNA-PKcs in UUO kidney tissues (Fig. 3a). The degree of tubular atrophy and dilatation induced by UUO was greatly improved after treatment with NU7441 compared to the vehicle control, as shown by PAS staining (Fig. 3b). Histopathological analyses of Masson staining revealed an approximately 50% reduction in renal interstitial fibrosis in the obstructed (UUO) kidneys of NU7441-treated mice compared to vehicle control UUO kidneys (Fig. 3c). Images of immunofluorescence staining showed that the deposition of interstitial collagen I induced by UUO was markedly reduced by NU7441 treatment (Fig. 3e). The protein levels of profibrotic markers, including FN and α-SMA, analyzed by Western blot were also significantly reduced in the UUO kidneys of NU7441-treated mice compared to vehicle controls (Fig. 3f). QRT-PCR analysis showed that NU7441 treatment markedly reduced expression levels of crucial kidney fibrosis genes, including *Col1a1, Col3a1, Fn1,* and *Acta2,* in kidney tissues of UUO mice compared to vehicle controls

(Supplementary Fig. 4a). Additionally, the infiltration of macrophages in kidney tissues of UUO mice was greatly reduced after treatment with NU7441 compared to vehicle controls (Fig. 3d). NU7441 also decreased expression levels of inflammation-related markers, including the cytokines *Il1β, Il6, Tnfα,* and *Mcp1*, in kidney tissues of UUO mice, as analyzed by qRT-PCR (Supplementary Fig. 4a). Furthermore, DNA-PKcs knockout mice were also treated with NU7441 to examine the possible off-target of NU7441. Our results showed NU7441 had no obvious anti-fibrotic effect in DNA-PKcs knockout mice (Fig. 3g, h), suggesting a specific action of NU7441 on DNA-PKcs in this experimental setting.

The protective effect of NU7441 against UUO-induced renal injury and renal fibrosis was extended to UIR-induced CKD. As expected, NU7441 treatment also protected against UIR-induced renal injury and fibrosis compared to vehicle controls (Supplementary Fig. 4b–h). Together, these results indicate that inhibition of DNA-PKcs activity by NU7441, attenuates CKD progression in mice.

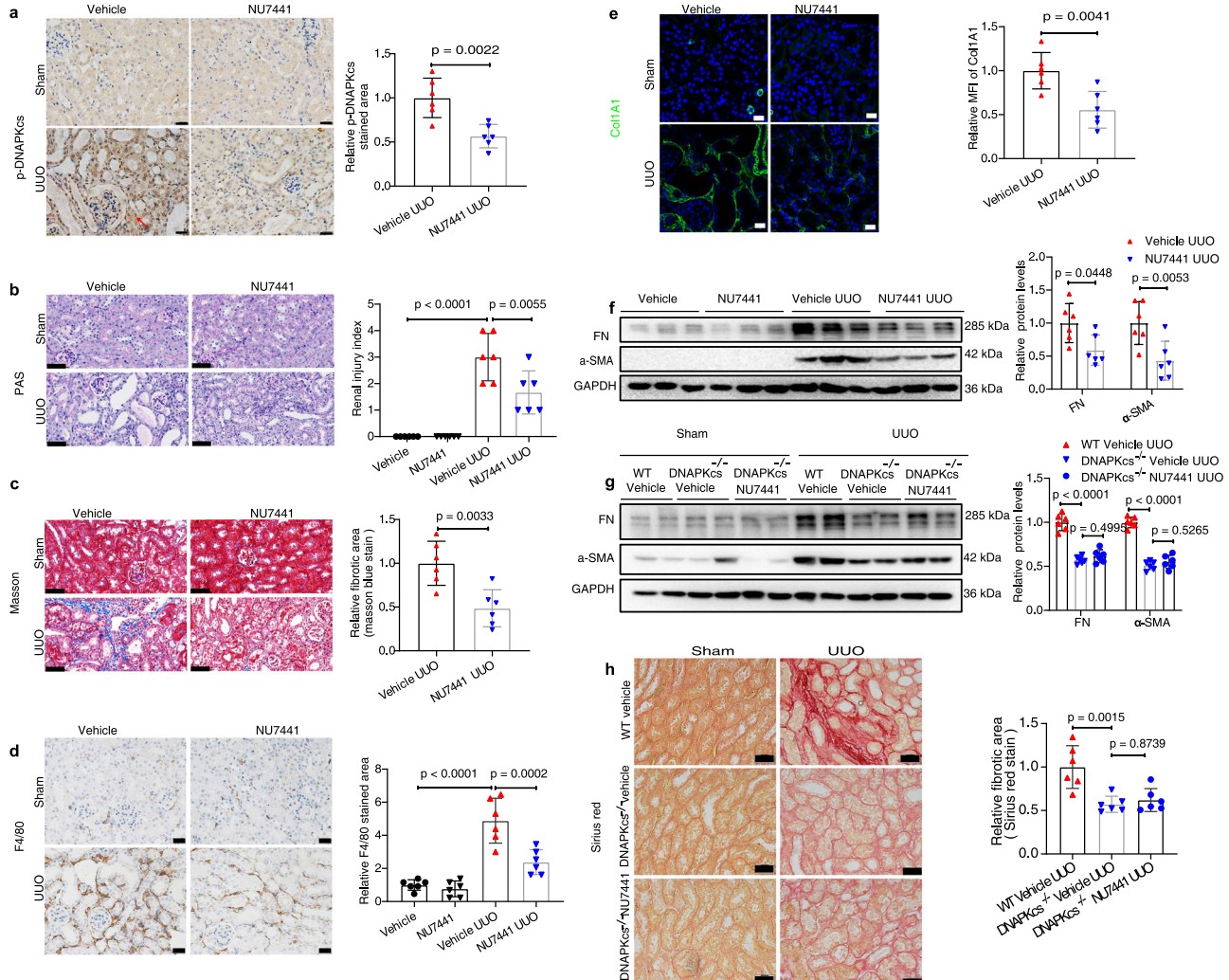

**Fig. 3 | Inhibition of DNA-PKcs activity attenuates the development of CKD in UUO mice. a** Immunohistochemical staining of p-DNA-PKcs in kidneys of mice treated with NU7441 (40 mg/kg) or vehicle subjected to UUO (day 7), scale bars: 20 μm. Bars represent quantification results (mean ± SD, $n = 6$ mice of each group). **b** PAS staining of kidneys of mice treated with NU7441 or vehicle subjected to UUO, scale bars: 50 μm. Bars represent quantification results (mean ± SD, $n = 6$ mice of each group). **c** Masson staining of kidneys of mice treated with NU7441 or vehicle subjected to UUO, scale bars: 50 μm. Bars represent quantification results (mean ± SD, $n = 6$ mice of each group). **d** Immunohistochemical staining of F4/80 in kidneys of mice treated with NU7441 or vehicle subjected to UUO, scale bars: 20 μm. Bars represent quantification results (mean ± SD, $n = 6$ mice of each group). **e** Immunofluorescence staining of COL1A1 in kidneys of mice treated with NU7441 or vehicle subjected to UUO, scale bars: 20 μm. Bars represent quantification results (mean ± SD, $n = 6$ mice of each group). **f** Protein levels of FN and α-SMA in kidneys of mice treated with NU7441 or vehicle subjected to UUO. Bars represent quantification results (mean ± SD, $n = 6$ mice of each group). **g** Protein levels of FN and α-SMA in kidneys of DNA-PKcs$^{-/-}$ or WT mice treated with NU7441 or vehicle subjected to UUO. Bars represent quantification results (mean ± SD, $n = 6$ mice of each group). **h** Sirius red staining of kidneys of DNA-PKcs$^{-/-}$ or WT mice treated with NU7441 or vehicle subjected to UUO. Bars represent quantification results (mean ± SD, $n = 6$ mice of each group). Two-tailed unpaired *t*-test was used to determine the *p*-values for **a**, **c**, **e**. One-way ANOVA followed by Tukey's multiple comparisons test was used to determine the *p*-values for **b**, **d**, **h**. Two-way ANOVAs followed by Šídák's multiple comparisons test were used to determine the *p*-values for **f**, **g**. Source data are provided as a Source Data file.

## Inhibition of DNA-PKcs preserves the tubular epithelial cell phenotype and regulates interstitial fibroblast activation in vitro

Epithelial injury and myofibroblast activation are central events in the pathogenesis of CKD[2]. Immunostaining revealed that p-DNAPKcs were obviously induced by TGFβ1 stimulation in both HK-2 cells (Fig. 4a) and primary tubular epithelial cells (Supplementary Fig. 5a). Western blot results showed that the phosphorylation and total protein levels of DNA-PKcs were also significantly upregulated in TGFβ1 or H₂O₂ treated HK-2 cells (Fig. 4i and Supplementary Fig. 5b). Additionally, DNA-PKcs knockout HK-2 cells were generated using CRISPR−Cas9, and WT and DNA-PKcs$^{-/-}$ mouse primary tubular epithelial cells were cultured. Successful construction of DNA-PKcs knockout HK-2 cells was confirmed by Western blotting (Supplementary Fig. 5c). Our results

showed that knockout of DNA-PKcs or inhibition of DNA-PKcs by NU7441 ameliorated the dedifferentiation of renal epithelial cells, as shown by lower levels of FN and Col1A1 production after exposure to TGFβ1 in DNA-PKcs knockout or NU7441-treated cells (Fig. 4b−d, k, l). However, KU80 knockout boosted FN production induced by TGFβ1 in tubular epithelial cells (Fig. 4m). Like UUO model, DNA-PKcs knockout didn't aggravate DNA breaks in HK-2 cells induced by H₂O₂ (Supplementary Fig. 5d). KU70/80 first recognizes DNA broken ends at DSB then recruits DNA-PKcs to form DNA-PK which is necessary in the repair of DSB via NHEJ[37]. These results suggested that DNA-PKcs enhanced the profibrotic action of renal epithelial cells in CKD independent of its DSB repair function.

To investigate whether DNA-PKcs is involved in the activation and proliferation of interstitial fibroblasts, NRK-49F rat renal fibroblasts

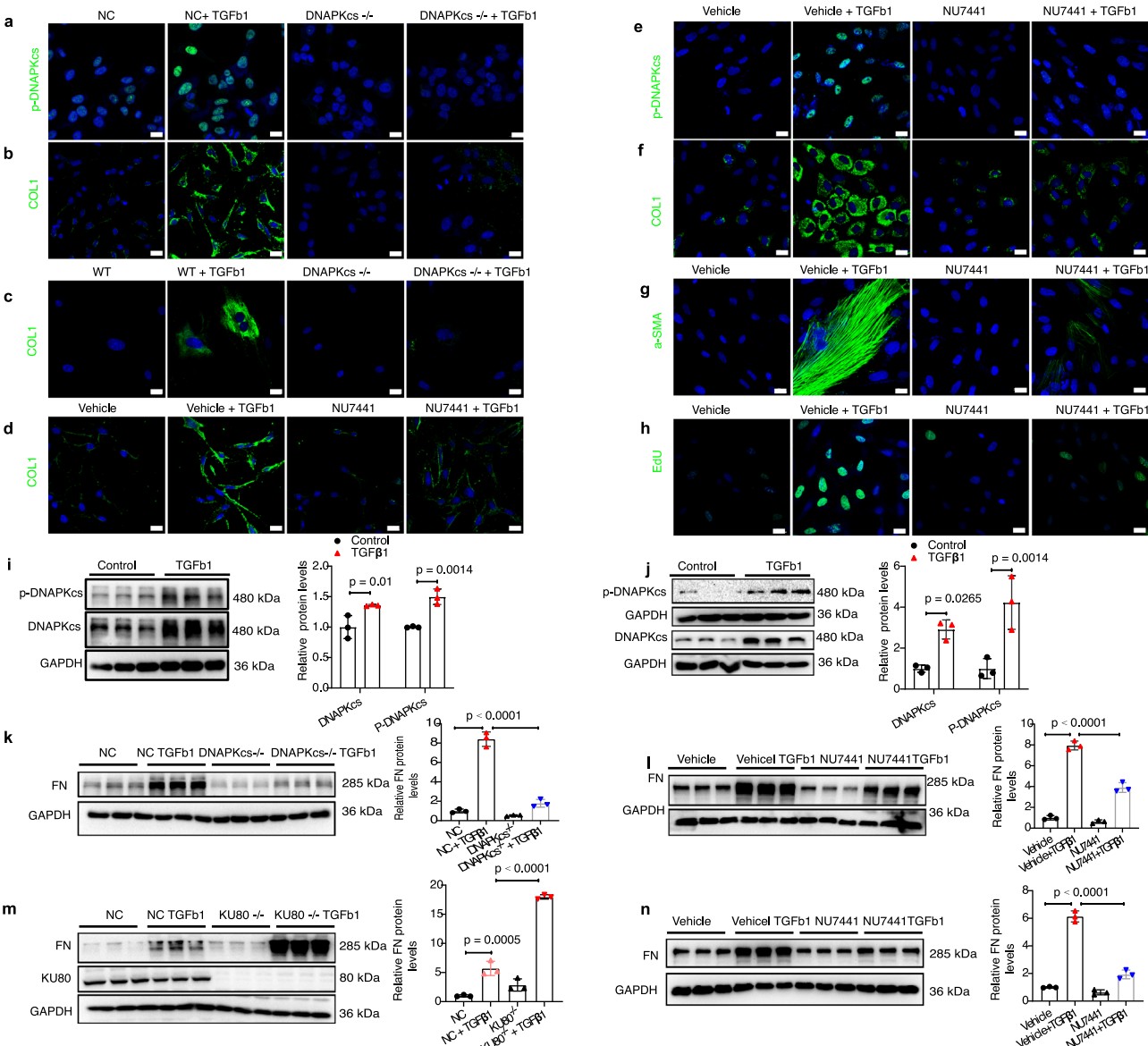

**Fig. 4 | Blockade of DNA-PKcs abolishes TGFβ1-induced tubular epithelial cell dedifferentiation and fibroblast activation. a** Representative immuno-fluorescence images of p-DNA-PKcs in DNA-PKcs$^{-/-}$ and NC (negative control) HK-2 cells treated with or without TGFβ1 (5 ng/ml) for 24 h, scale bars: 20 μm, $n = 3$ biologically independent experiments. **b** Representative immunofluorescence images of COL1A1 in HK-2 cells, primary tubular epithelial cells (**c**) and NU7441 (0.1 μM) or vehicle-treated HK-2 cells (**d**) treated with or without TGFβ1 (5 ng/ml) for 24 h, scale bars: 20 μm. $n = 3$ biologically independent experiments. **e** Representative immunofluorescence images of p-DNA-PKcs, COL1 (**f**), α-SMA (**g**) and EdU (**h**) in NU7441 (0.1 μM) or vehicle-treated NRK-49F cells treated with or without TGFβ1 (5 ng/ml) for 24 h, scale bars: 20 μm, $n = 3$ biologically independent experiments. **i** Protein levels of p-DNA-PKcs and DNA-PKcs in HK-2 cells treated with or without TGFβ1 (5 ng/ml) for 24 h. Bars represent quantification results (mean ± SD, $n = 3$ biologically independent experiments). **j** Protein levels of p-DNA-PKcs and

DNA-PKcs in NRK-49F cells treated with or without TGFβ1 (5 ng/ml) for 24 h. Bars represent quantification results (mean ± SD, $n = 3$ biologically independent experiments). **k** Protein levels of FN in DNA-PKcs$^{-/-}$ or NU7441-treated (**l**) HK-2 cells analyzed by Western blot. Bars represent quantification results (mean ± SD, $n = 3$ biologically independent experiments). **m** Protein levels of FN and KU80 in KU80$^{-/-}$ and NC HK-2 cells treated with or without TGFβ1 (5 ng/ml) for 24 h. Bars represent quantification results (mean ± SD, $n = 3$ biologically independent experiments). **n** Western blot analysis of protein levels of FN in NU7441 (0.1 μM) or vehicle-treated NRK-49F cells. Bars represent quantification results (mean ± SD, $n = 3$ biologically independent experiments). One-way ANOVA followed by Tukey's multiple comparisons test was used to determine the $p$-values for **k**–**n**. Two-way ANOVAs followed by Šídák's multiple comparisons test were used to determine the $p$-values for **i**, **j**. NC negative control. Source data are provided as a Source Data file.

were cultured. Western blot results showed that the phosphorylated and total protein levels of DNA-PKcs were also significantly upregulated in TGFβ1 treated NRK-49F cells (Fig. 4j). Immunostaining revealed that the activity of DNA-PKcs was markedly induced by TGFβ1 treatment in NRK-49F cells (Fig. 4e) but was markedly inhibited by NU7441 treatment. Treatment of NRK-49F cells with TGFβ1 induced fibroblast activation and myofibroblast differentiation with upregulation of COL1, α-SMA and FN, whereas pretreatment with NU7441

greatly blunted the profibrotic phenotype (Fig. 4f, g, n). Moreover, NU7441 treatment significantly inhibited NRK-49F cell proliferation induced by TGFβ1 compared to vehicle controls, as indicated by EdU staining (Fig. 4h). Taken together, these data indicate that inhibition of DNA-PKcs preserves the tubular epithelial cell phenotype and regulates interstitial fibroblast activation induced by TGFβ1 in vitro.

Moreover, we also examined the mechanism of TGFβ1-induced DNA-PKcs expression. First, we found multiple possible binding sites

of SMAD2 in the promoter region of DNA-PKcs through JASPAR transcription factor binding profiles analysis[38]. Second, luciferase assay results showed that overexpression of SMAD2 increased transcriptional activation of DNA-PKcs (Supplementary Fig. 5e). Additionally, the effect of DNA-PKcs on the activation of SMAD signaling was also analyzed. The results showed DNA-PKcs knockout inhibited the activation of SMAD2/SMAD3 in UUO mouse model (Supplementary Fig. 5f). These results suggested SMAD2/SMAD3 upregulated DNA-PKcs expression, while DNA-PKcs activated SMAD2/SMAD3 (Supplementary Fig. 5g), forming a positive loop.

## DNA-PKcs-mediated phosphorylation of TAF7 aggravates renal fibrosis

To further dissect the possible molecular mechanisms by which DNA-PKcs regulates the development of CKD, phosphoproteomics was performed to reveal possible substrates of DNA-PKcs. As shown in Fig. 5a, most of the differentially phosphorylated proteins were nuclear proteins. The phosphorylation levels of only 6 proteins of these proteins were significantly decreased in kidney tissues of DNA-PKcs[−/−] mice compared to WT mice, as shown by the volcano plots (Fig. 5a). Heatmap analysis of differentially phosphorylated proteins between DNA-PKcs[−/−] mice and WT controls showed that phosphorylation levels of the indicated sites of Slc4a1, Ampd2, Hbb-b1, Gas2, Ank1 and TAF7 were significantly decreased after DNA-PK knockout (Fig. 5a). Through subcellular localization analysis and knockdown of these proteins, we found that DNA-PKcs mediates renal fibrosis, perhaps through phosphorylation of TAF7, which is a subunit of TFIID. Coimmunoprecipitation studies revealed a possible direct interaction between DNA-PKcs and TAF7 (Fig. 5b). Protein-Protein docking between DNA-PKcs and TAF7 was analyzed through an online database ClusPro[39,40]. The docking results showed a possible direct interaction between DNA-PKcs and TAF7, and the 213-site serine of TAF7 is one of the nearest sites perhaps binding to DNA-PKcs (Supplementary Fig. 6a). An in vitro kinase assay was performed to analyze whether TAF7 was phosphorylated by DNA-PKcs directly. The products of kinase assay were analyzed by mass spectrometry. The results showed the S213, T137, Y24, and S171 sites of human TAF7 were phosphorylated by DNA-PKcs (Supplementary Fig. 6b). Furthermore, our results revealed a positive association between the protein levels of p-DNA-PKcs and TAF7 in human CKD kidney tissues via immunofluorescence co-staining of p-DNA-PKcs with TAF7 (Fig. 5c).

A previous study found the calculated molecular weight of TAF7 is about 40 kDa, but phosphorylated TAF7 is approximately 55 kDa[41]. Western blot and immunohistochemical staining results showed that levels of TAF7 were greatly induced by UUO in the kidney tissues of WT controls (Fig. 5d and Supplementary Fig. 6d). DNA-PKcs knockout inhibited the phosphorylation of TAF7 in both the baseline and UUO models compared to WT controls (Fig. 5d). The phosphorylation of TAF7 in UUO mice was also confirmed by Phos-tag SDS-PAGE[42] (Supplementary Fig. 6c). Similar results were also observed in UIR model (Fig. 5e). Treatment with the DNA-PKcs inhibitor NU7441 also inhibited the upregulated phosphorylation of TAF7 induced by UIR in mice (Fig. 5f). Additionally, protein levels of TAF7 analyzed by Western blot showed that both DNA-PKcs knockout in HK-2 cells (Fig. 5g) and treatment with NU7441 in HK2 (Fig. 5h) or NRF-49F cells (Fig. 5i) inhibited the upregulated phosphorylation of TAF7 induced by TGFβ1 in vitro.

Since DNA-PKcs is involved in the phosphorylation of TAF7 in renal fibrosis, we examined whether TAF7 deficiency may affect the dedifferentiation of renal epithelial cells or the activation and proliferation of interstitial fibroblasts induced by TGFβ1. TAF7 knockout mPTC cells and NRK-49F cells were generated using CRISPR/Cas9, and successful construction of TAF7 knockout cells was confirmed by Western blot (Supplementary Fig. 6e). Our results showed that TAF7 knockout ameliorated the dedifferentiation of renal epithelial cells, as

shown by low levels of FN and collagen I production after exposure to TGFβ1 (Fig. 5j, l). Similarly, TAF7 knockout blunted the profibrotic phenotype of NRF-49F cells induced by TGFβ1, as shown by low levels of FN production (Fig. 5k), collagen I (Fig. 5m), α-SMA (Fig. 5n) as well as low levels of EdU staining (Fig. 5o) in TAF7[−/−] cells. To determine whether overexpression of TAF7 in renal epithelial cells is sufficient to drive a profibrotic phenotype, HK-2 cells or NRK49F cells were transfected with a human TAF7 or its mutant's overexpression plasmids. Western blot results showed that the profibrotic phenotype of HK-2 or NRF49F cells was indeed enhanced by overexpression of TAF7 or its mutants (Supplementary Fig. 6h,j). However, the profibrotic effects of TAF7 were almost entirely blocked by DNA-PKcs knockout in HK-2 cells (Supplementary Fig. 6i). Furthermore, we examined whether TAF7 deficiency may affect the profibrotic effects of DNA-PKcs. Interestingly, TAF7 deficiency inhibited the profibrotic effects of DNA-PKcs overexpression (Supplementary Fig. 6f) and blocked the antiprofibrotic effects of NU7441 in mPTC cells (Supplementary Fig. 6g). Recently, Wang S, et al. found cytoplasmic DNA-PKcs were increased and interacted with Fis1 and phosphorylated it at Thr34 in its TQ motif, which increased the affinity of Fis1 for Drp1 and induced mitochondrial fragmentation in AKI[43]. In our study, nuclear extraction protein analysis results showed DNA-PKcs was mainly localized and increased in the nucleus in kidneys of UUO mice which was different from AKI model (Supplementary Fig. 6k). And knocking down Fis1 promoted the profibrotic response in renal tubular epithelial cells challenged with TGFβ1, suggesting that Fis1 did not mediate the profibrotic role of DNA-PKcs in CKD (Supplementary Fig. 6l). Taken together, these data indicated that DNA-PK mediated phosphorylation of TAF7 aggravates renal fibrosis.

## TAF7 promotes mTORC1 activation by upregulating expression of RAPTOR

To investigate the mechanisms of the profibrotic effects of TAF7, RNA-Seq was performed to analyze transcriptomic changes in NC and TAF[−/−] mPTC cells treated with or without TGFβ1. RNA-Seq results showed that TAF7 knockout ameliorated TGFβ1-induced fibrosis-associated gene expression in mPTCs (Supplementary Fig. 7a, b). KEGG pathway enrichment analysis showed that the mTOR signaling pathway was the top pathway between TAF[−/−] and NC mPTC cells without TGFβ1 treatment (Fig. 6a). Gene set enrichment analysis (GSEA) revealed a significant upregulation of the mTOR signaling pathway after treatment with TGFβ1, which was significantly decreased after TAF7 knockout in mPTC cells compared to NC control cells (Fig. 6a). Heatmap analysis of mTOR signaling demonstrated that RPTOR (RAPTOR), a positive regulator of mTORC1[44,45], was increased by treatment with TGFβ1, which was inhibited by knockout of TAF7 in mPTC cells (Fig. 6b).

Protein levels of RPTOR and phosphorylation of mTOR were both significantly increased after treatment with TGFβ1 in mPTC cells as analyzed by Western blot. However, TAF7 deficiency decreased the protein levels of RPTOR and the phosphorylation of mTOR induced by TGFβ1 in mPTC cells compared to the NC controls (Fig. 6c). MRNA levels of mTOR and its protein partners, including *Raptor, Rictor,* and *Mlst8*, were analyzed by qRT-PCR, and the results showed that TGFβ1 treatment upregulated the mRNA levels of *Rptor, Rictor*, and *Mlst8* but not *mTOR*. TAF7 deficiency significantly decreased the mRNA levels of *Raptor* both with and without TGFβ1 treatment in mPTC cells compared to NC control cells but not *Rictor, Mlst8,* or *mTOR* (Supplementary Fig. 7c). Similarly, DNA-PKcs deficiency or inhibition significantly decreased mRNA levels of *Raptor* in kidney tissues of UUO mice and UIR mice (Supplementary Fig. 7d, e). Additionally, we constructed two different residue 213 serine site mutants of TAF7, TAF7-S213D, and TAF7-S213A to analyze the phosphorylation of serine 213 of TAF7 on transcriptional activation of human *Raptor* induced by TGFβ1 in HK-2 cells. The luciferase assay results showed that overexpression of all these TAF7 mutants increased transcriptional activation of

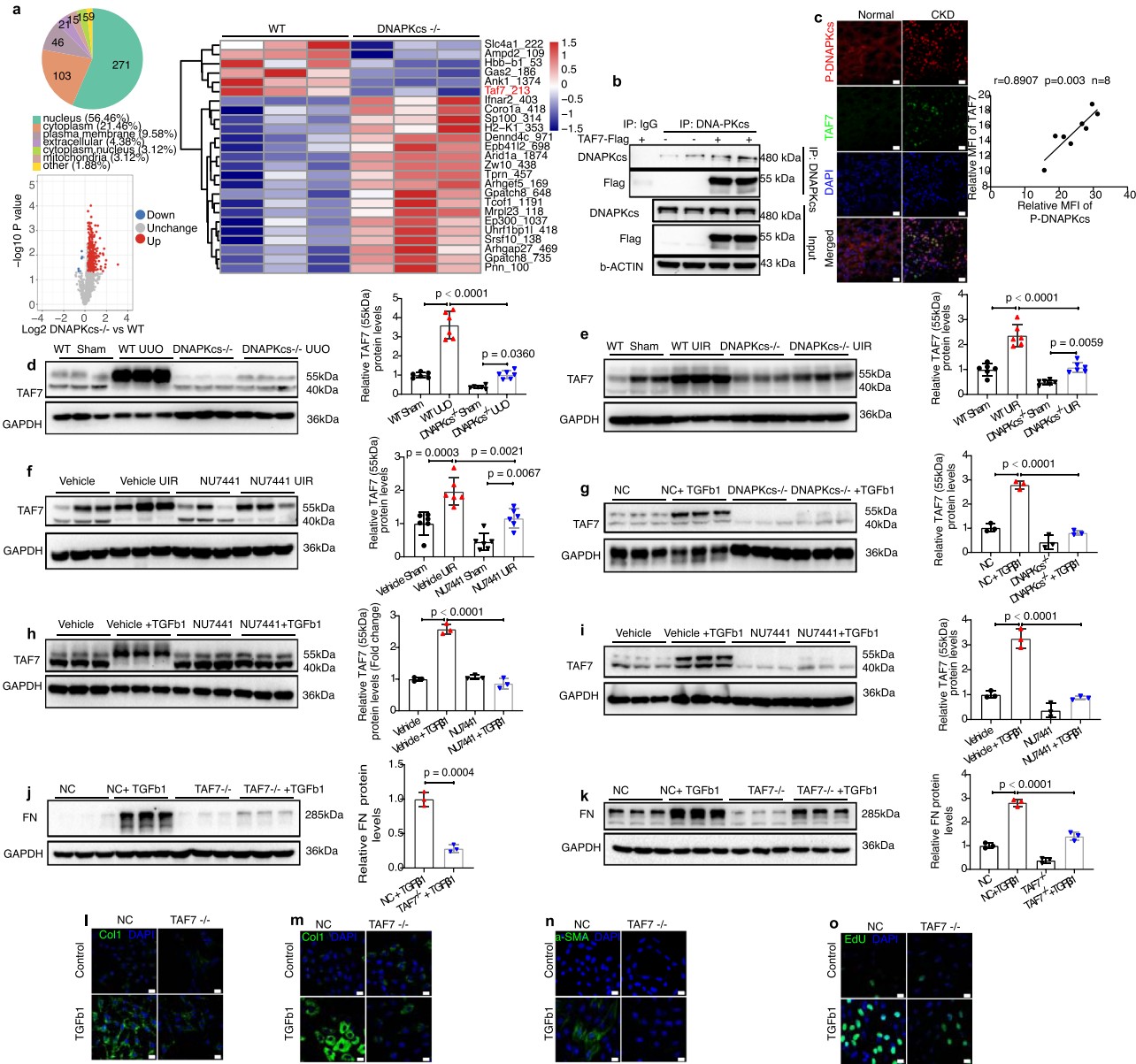

**Fig. 5 | DNA-PKcs-mediated phosphorylation of TAF7 aggravates renal fibrosis.**
**a** Phosphoproteomics profiling of WT and DNA-PKcs$^{-/-}$ kidney tissues. Differentially expressed statistical analysis, including subcellular localization, volcano plot and heat-map analysis, is shown ($n = 3$ mice of each group). **b** the interaction between Flag-tagged TAF7 and endogenous DNA-PKcs visualized by immunoprecipitation with an anti-DNA-PKcs antibody and immunoblotting with an anti-Flag antibody ($n = 3$ independent experiments). **c** Immunofluorescence co-staining of p-DNA-PKcs with TAF7 in kidney tissues of normal and CKD human study participants, red: p-DNA-PKcs, green: TAF7, bule: DAPI, scale bars: 20 μm. Pearson's r correlation analysis between TAF7 and p-DNA-PKcs protein levels in CKD patients ($n = 8$) with 95% confidence interval from 0.4997 to 0.980. **d** Protein levels of TAF7 in DNA-PKcs $^{-/-}$ or WT mice subjected to UUO (day 7) or UIR (day 21) (**e**); NU7441 (40 mg/kg) or vehicle treatment mice subjected to UIR (**f**), bars represent quantification results (mean ± SD, $n = 6$ mice); DNA-PKcs$^{-/-}$ HK-2 cells treated with TGFβ1 (5 ng/ml) for

24 h (**g**) and NU7441 (0.1 μM) treatment HK2 (**h**) or NRK-49F(**i**) cells treated with TGFβ1 (5 ng/ml) for 24 h. Bars represent quantification results (mean ± SD, $n = 3$ independent experiments). **j** Protein levels of FN in TAF7$^{-/-}$ and NC mPTCs treated with TGFβ1 (5 ng/ml) for 24 h. Bars represent quantification results (mean ± SD, $n = 3$ independent experiments). **k** Protein levels of FN in TAF7$^{-/-}$ and NC NRK-49F treated with TGFβ1 (5 ng/ml) for 24 h. Bars represent quantification results (mean ± SD, $n = 3$ independent experiments). **l** Immunofluorescence images of COL1A1 in TAF7$^{-/-}$ and NC mPTCs treated with TGFβ1 (5 ng/ml) for 24 h, scale bars: 20 μm ($n = 3$ independent experiments). **m** Immunofluorescence images of COL1A1, α-SMA (**n**) and EdU (**o**) in TAF7$^{-/-}$ and NC NRK-49F cells treated with TGFβ1 (5 ng/ml) for 24 h, scale bars: 20 μm ($n = 3$ biologically independent experiments). Two-tailed unpaired $t$-test was used to determine the $p$-values for **j**. One-way ANOVA followed by Tukey's multiple comparisons test was used to determine the $p$-values for **d–i**, **k**. Source data are provided as a Source Data file.

human *Raptor*, although the relative luciferase activity of the TAF7-S213A group was lower than that of the TAF7-WT and TAF7-S213D groups (Fig. 6d). Moreover, ChIP assay results showed that TAF7 could bind to *Rptor* promoter directly (Fig. 6e). These results indicated that TAF7 mediates transcriptional activation of RPTOR induced by TGFβ1, perhaps not only through phosphorylation of residue 213. Furthermore, our results showed that protein levels of RPTOR and

phosphorylation of mTOR were both increased in kidney tissues of UUO and UIR mice and were significantly decreased after DNA-PKcs knockout in both baseline and model mice (Fig. 6f–i). Similar results were also observed in response to DNA-PKcs knockout HK-2 cells and NU7441 treatment in NRF-49F cells (Fig. 6j, k). Additionally, Western blot results showed that protein levels of RPTOR and phosphorylation of mTOR were increased after overexpression of either DNA-PKcs or

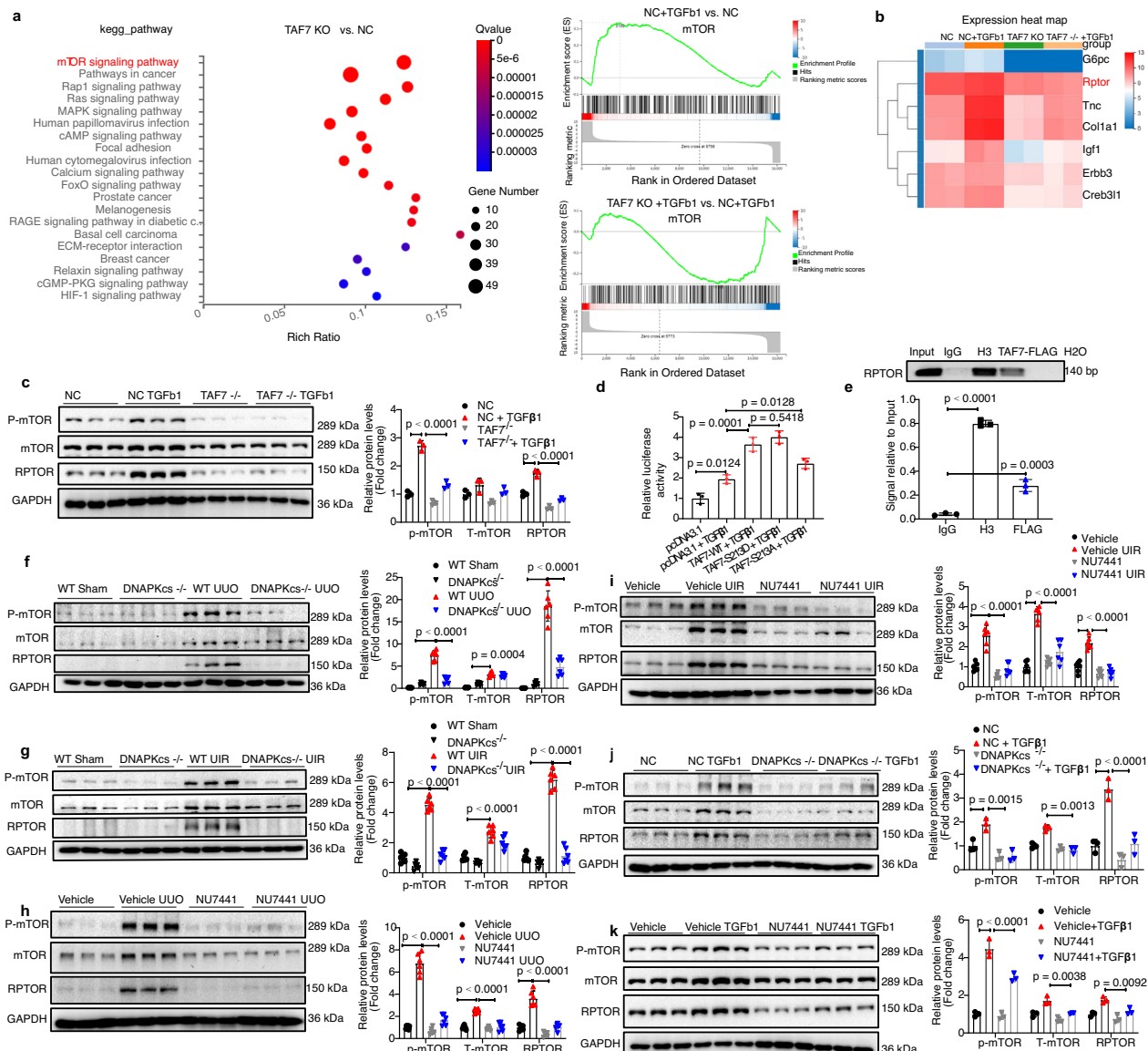

**Fig. 6 | DNA-PKcs-mediated phosphorylation of TAF7 promotes mTORC1 activation by upregulating RPTOR expression. a** RNA-seq showing that mTOR signaling (indicated by red font) was the top differentially expressed signaling factor between the TAF7$^{-/-}$ and NC groups. GSEA showing that mTOR signaling was upregulated in the NC + TGFβ1 group and suppressed in the TAF7$^{-/-}$ + TGFβ1 group (n = 2), TGFβ1 (5 ng/ml) treatment for 24 h. **b** Heatmap image showing that *Rptor* (indicated by red font) was decreased in the TAF7$^{-/-}$ groups compared to the NC groups with or without TGFβ1 (5 ng/ml) treatment for 24 h (n = 2 of each group). **c** Protein levels of RPTOR, mTOR and phosphorylated mTOR analyzed by Western blot in TAF7$^{-/-}$ and NC mPTCs treated with TGFβ1 (5 ng/ml) for 24 h. Bars represent quantification results (mean ± SD, n = 3 independent experiments). **d** Luciferase reporter assay in HK-2 cells with TGFβ1 (5 ng/ml) for 24 h. Bars represent results of 3 independent experiments (mean ± SD). **e** ChIP assay was performed to analysis

TAF7 binds to the promoter of *Rptor*. Bars represent quantification results (mean ± SD, n = 3 independent experiments). **f** Protein levels of RPTOR, mTOR and phosphorylated-mTOR were analyzed by Western blot in kidney tissues of DNA-PKcs$^{-/-}$ and WT control mice subjected to UUO (day 7) or UIR (day 21) (**g**), NU7441 (40 mg/kg) and vehicle treatment mice subjected to UUO (**h**) or UIR (**i**), in DNA-PKcs$^{-/-}$ or NC HK-2 cells (**j**), NU7441 (0.1 μM) or vehicle treatment NRK-49F cells (**k**) treated with TGFβ1 (5 ng/ml) for 24 h. Bars represent quantification results (mean ± SD, n = 6 mice of each group and n = 3 independent experiments for cell models). One-way ANOVA followed by Tukey's multiple comparisons test was used to determine the *p*-values for **d**, **e**. Two-way ANOVAs followed by Šídák's multiple comparisons test were used to determine the *p*-values for **c**, **f**–**k**. Source data are provided as a Source Data file.

TAF7 (Supplementary Fig. 7f,g). Moreover, knockdown of RPTOR markedly inhibited the phosphorylation of mTOR induced by TGFβ1 (Supplementary Fig. 7h). Taken together, these results indicate that DNA-PKcs-mediated phosphorylation of TAF7 promotes mTORC1 activation by upregulating the expression of RPTOR.

## Inhibition of DNA-PK corrects metabolic reprogramming in vivo and in vitro

According to several previous studies[9,46], aberrant activation of mTORC1 promotes the progression of renal fibrosis by mediating

metabolic reprogramming in fibrotic kidneys. Thus, we examined whether inhibition of DNA-PK or TAF7 corrects metabolic reprogramming in fibrotic kidneys. Based on RNA-Seq analysis of transcriptomic changes in TGFβ1-treated NC and TAF$^{-/-}$ mPTC cells, KEGG pathway enrichment analysis showed that the metabolic signaling pathway was one of the most differentially expressed pathways between TGFβ1-treated TAF$^{-/-}$ and NC mPTC cells (Fig. 7a). Furthermore, GSEA revealed significant upregulation of oxidative phosphorylation and the fatty acid oxidation (FAO) pathway after treatment with TGFβ1 in TAF7 knockout mPTC cells compared to NC control cells

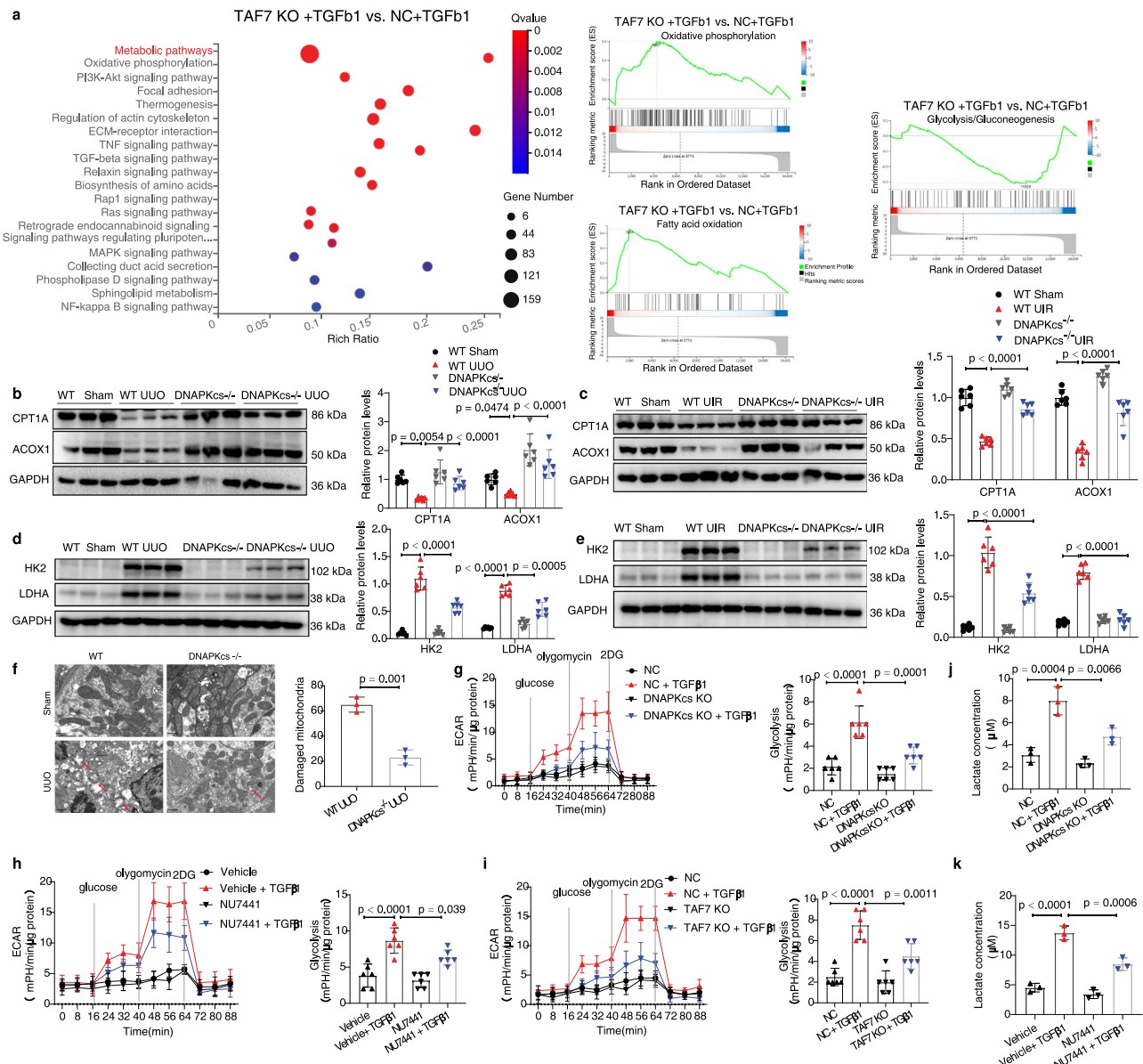

**Fig. 7 | Inhibition of DNA-PK or TAF7 corrects metabolic reprogramming.**
**a** RNA-seq showing that metabolic signaling (indicated by red font) was the top differentially expressed signaling pathway between TAF7$^{-/-}$ + TGFβ1 and NC + TGFβ1. GSEA revealed significant upregulation of oxidative phosphorylation and FAO but inhibition of glycolysis in TGFβ1-treated TAF7 knockout mPTCs compared to NC controls, $n = 2$ for each group. **b** Protein levels of ACOX1 and CPT1α were analyzed in kidney tissues of DNA-PKcs$^{-/-}$ and WT control mice subjected to UUO (day 7) or UIR (day 21) (**c**). Bars represent quantification results (mean ± SD, $n = 6$ mice of each group). **d** Protein levels of LDHA and HK2 were analyzed in kidney tissues of DNA-PKcs$^{-/-}$ and WT control mice subjected to UUO (day 7) or UIR (day 21) (**e**). Bars represent quantification results (mean ± SD, $n = 6$ mice of each group). **f** Electron microscopy showing that UUO (day 7)-induced mitochondrial damage in proximal tubule cells of WT mice was almost eliminated by DNA-PKcs knockout, scale bars: 1 μm. Bars represent quantification results (mean ± SD, $n = 3$ mice of each

group). **g** ECAR of TGFβ1-treated (5 ng/ml, 24 h) DNA-PKcs$^{-/-}$ HK-2 cells, NU7441 (0.1 μM)-treated NRK-49F cells (**h**) and TAF7$^{-/-}$ mPTCs (**i**) were analyzed using Seahorse XF96 cell culture microplates to record baseline (before glucose), glycolysis rate (after glucose), glycolytic capacity (after oligomycin), and glycolytic reserve (after 2-DG). Bars represent the rate of glycolysis (mean ± SD, $n = 6$ sample of each group). **j** Lactate production in the medium of DNA-PKcs$^{-/-}$ HK-2 cells and NU7441 (0.1 μM)-treated NRK-49F cells (**k**) treated with TGFβ1 (5 ng/ml) for 24 h. Bars represent quantification results (mean ± SD, $n = 3$ independent experiments). Two-tailed unpaired t-test was used to determine the $p$-values for **f**. One-way ANOVA followed by Tukey's multiple comparisons test was used to determine the $p$-values for **g–k**. Two-way ANOVAs followed by Šídák's multiple comparisons test were used to determine the $p$-values for **b–e**. Source data are provided as a Source Data file.

(Fig. 7a). The glycolysis pathway was significantly decreased when TAF7 was knocked out in mPTCs compared to the NC control (Fig. 7a). These results indicated that TAF7 deficiency corrects TGFβ1-induced metabolic reprogramming in mPTC cells.

To understand the role of DNA-PKcs in regulating metabolic reprogramming in fibrotic kidneys. First, the protein and mRNA levels of key enzymes involved in fatty acid oxidation (CPT1, CPT2, ACOX1, and ACOX2) were analyzed by Western blot and qRT-PCR. The results

showed that expression of these enzymes was significantly reduced in the kidneys of WT mice under UUO or UIR conditions, but their expression was significantly restored in DNA-PKcs knockout mice (Fig. 7b, c and Supplementary Fig. 8). Second, mitochondrial morphology and function are necessary for oxidative phosphorylation, and electron microscopy revealed obvious fragmentation of the mitochondria in kidney tubule cells of WT mice induced by UUO, which was significantly improved in kidney tubule cells of DNA-PKcs$^{-/-}$

mice (Fig. 7f). These results revealed that the abnormal oxidative phosphorylation and fatty acid metabolism induced by UUO in WT mice is improved by DNA-PKcs knockout.

Glycolysis is a feature of renal fibrosis and promotes the development of renal fibrosis. Protein levels of hexokinase 2 (HK2) and lactate dehydrogenase A (LDHA), which are rate-limiting enzymes for glycolysis, were analyzed by Western blot. The results showed that levels of HK2 and LDHA were significantly upregulated in kidney tissues of UUO mice and UIR mice, indicating increased glycolytic activity in fibrotic kidneys. In contrast, glycolysis was significantly inhibited in kidney tissues of DNA-PKcs-deficient mice compared to WT mice in both the UUO and UIR models, as indicated by lower levels of HK2 and LDHA (Fig. 7d, e). Extracellular acidification rate (ECAR) assays were performed to analyze glycolytic activity in TGFβ1-treated DNA-PKcs$^{-/-}$ HK-2 cells and NU7441-treated NRK-49F cells in vitro. Both knockout of DNA-PKcs and treatment with NU7441 significantly decreased the ECAR, which was upregulated in TGFβ1-treated cells compared to control cells (Fig. 7g–h). Knockout of TAF7$^{-/-}$ also significantly decreased the upregulation of ECAR in mPTCs induced by TGFβ1 (Fig. 7i). Additionally, extracellular lactate levels in the culture medium of DNA-PKcs$^{-/-}$ and NU7441-treated cells were also decreased (Fig. 7j, k).

To understand how metabolic reprogramming was improved in DNA-PKcs$^{-/-}$ mice, we used metabolomics profiling to analyze metabolite changes following UUO surgery (Fig. 8a). UUO induced impaired glucose metabolism in the kidneys of WT mice, as evidenced by the downregulation of glucose metabolites, such as pyruvate, in WT UUO kidneys, which was restored by knockout of DNA-PKcs (Fig. 8a). Additionally, increased accumulation of lactate in the kidneys of WT mice subjected to UUO was also observed, indicating a metabolic shift from oxidative phosphorylation to enhanced glycolysis in kidney cells. However, in the injured kidneys of DNA-PKcs$^{-/-}$ mice, glycolysis was significantly inhibited (Fig. 8a). Moreover, accumulation of carnitine-conjugated long-chain fatty acids, including lauroyl carnitine (Fig. 8a) and myristoyl carnitine (Fig. 8a), was observed in the kidneys of WT UUO mice, suggesting defects in fatty acid metabolism. In contrast, DNA-PKcs knockout largely corrected all these abnormalities induced by UUO. Taken together, these results indicate that inhibition of DNA-PK or TAF7 corrects metabolic reprogramming induced by injury or TGFβ1 in vivo and in vitro.

## Discussion

Epithelial dedifferentiation and myofibroblast activation and proliferation, all initiated by cellular injury, promote the progression of CKD[2]. Although the DNA damage response (DDR) often correlates with renal epithelial injury[47], little is known about its potential role in epithelial dedifferentiation and myofibroblast activation in progressive CKD. In this study, our results revealed that the expression of DNA-PKcs, a DNA-dependent protein kinase catalytic subunit, increased in fibrotic kidneys and promotes the development of CKD. Although DNA-PKcs is activated by DNA double-stranded breaks (DSBs)[24,25] or ROS[26], our results showed DNA-PKcs expression was also activated by TGFβ1-SMAD signaling, while DNA-PKcs knockout inhibited SMAD2/ SMAD3. These results suggest a new possible pathway mediating TGFβ1-SMAD signaling activation in fibrosis.

Next, we generated global prkdc gene-knockout mice. Our results revealed that deletion of DNA-PKcs attenuated renal tubular injury and the progression of renal interstitial fibrosis in both UUO and UIR mouse models. We cannot conclude that DNA-PK promotes renal fibrosis independent of lymphocytes, as the most prominent phenotype of DNA-PKcs$^{-/-}$ mice is lymphocyte deficiency[27,48]. To exclude the potential effect of lymphocyte deficiency, we constructed proximal renal tubular epithelial cell with specific DNA-PKcs knockout in vivo by using CRISPR/cas9 knock in mice. Our results revealed that renal tubular specific deletion of DNA-PKcs also hampered the progression

of renal interstitial fibrosis in UUO. Additionally, our results also showed that DNA-PKcs deficiency preserved the tubular epithelial cell phenotype and regulated interstitial fibroblast activation in vitro. These results indicate that DNA-PKcs mediates epithelial dedifferentiation and myofibroblast activation, probably without any direct relationship with lymphocyte deficiency. Additionally, we investigated the antifibrotic effects of a highly specific DNA-PKcs inhibitor, NU7441, and the results showed that NU7441 treatment markedly attenuated the progression of renal interstitial fibrosis in both UUO and UIR mouse models. NU7441 is a partial inhibitor of DNA-PK at physiological doses. A previous study indicated that NU7441 treatment did not significantly affect B cell function in adult mice[33]. Moreover, we confirmed the specific action of NU7441 on DNA-PKcs by treating DNA-PKcs knockout mice with NU7441 in UUO mouse model. These results suggest potential transformation of NU7441 for the clinical treatment of CKD, although further research is needed.

Our results showed DNA-PKcs deficiency did not aggravate DSBs in vivo and in vitro, suggesting that DNA-PKcs mediates renal injury and renal interstitial fibrosis, probably not through its role in DDR. Phosphoproteomics analysis showed that phosphorylation of TAF7 was decreased in the kidney tissues of DNA-PKcs$^{-/-}$ mice. TAF7 deficiency inhibited the profibrotic phenotype of renal epithelial cells and fibroblasts induced by TGFβ1. The profibrotic effects of TAF7 were almost entirely blocked by DNA-PKcs deficiency in renal epithelial cells. Our results indicate that TAF7 is a substrate for DNA-PKcs kinase activity and that DNA-PKcs-mediated phosphorylation of TAF7 aggravates renal fibrosis.

RNA-Seq results showed that TAF7 promotes mTORC1 activation by upregulating the expression of Raptor, which is a positive regulator of mTORC1[49,50]. Phosphoproteome results showed that phosphorylation of TAF7 on serine-213 was significantly decreased in the kidneys of DNA-PKcs$^{-/-}$ mice, compared to that in WT controls. Mass spectrometry analysis indicated the S213, T137, Y24, and S171 sites of human TAF7 were phosphorylated by DNA-PKcs in vitro. Moreover, transfection of different TAF7 mutants, including TAF7-WT, TAF7-S213D, and TAF7-S213A, increased transcriptional activation of human Rptor in HK-2 cells. These results indicate that TAF7 mediates transcriptional activation of Rptor induced by TGFβ1, probably not only through phosphorylation of its 213-site. TAF7 is not a core transcription factor but interacts with other TAFs, such as TAF1[41,51], and regulates the enzymatic activities of transcription factors, which could explain why overexpression of TAF7 alone did not activate the transcription of human RPTOR in HK-2 cells without TGFβ1 treatment. Several serine residues of TAF7 can be phosphorylated to activate or inhibit transcription[41]. For instance, a previous study found that phosphorylation of TAF7 on serine 264 disrupted TAF7 binding to TAF1, resulting in transcriptional upregulation of cyclin D1[41]. In addition to TAF1-dependent transcription, TAF7 also regulates TAF1-independent transcription[51]. In this study, ChIP assay results demonstrated that TAF7 could bind to Rptor promoter directly, suggesting TAF7 perhaps collaborates with other transcription factors to initiate Rptor expression. Although the mechanisms of TAF7-mediated transcriptional upregulation of Rptor need further research, our results indicate that DNA-PKcs mediates the phosphorylation of TAF7 to promote mTORC1 activation by upregulating the expression of Rptor.

Several previous studies have reported that mTORC1 signaling is activated in dedifferentiated epithelial cells and myofibroblasts from fibrotic kidneys[9,52]. Although DNA-PK has also been reported to phosphorylate and activate AKT/mTOR signaling in tumor cells in response to DNA breaks[53], the precise mechanism is unknown. Our results revealed that DNA-PKcs-mediated phosphorylation of TAF7 promotes mTORC1 activation by upregulating the expression of Raptor, providing a possible mechanism of mTORC1 activation in fibrotic kidneys. Chronic activation of mTORC1 promotes metabolic reprogramming in many diseases, including CKD[9,54]. Recent studies have

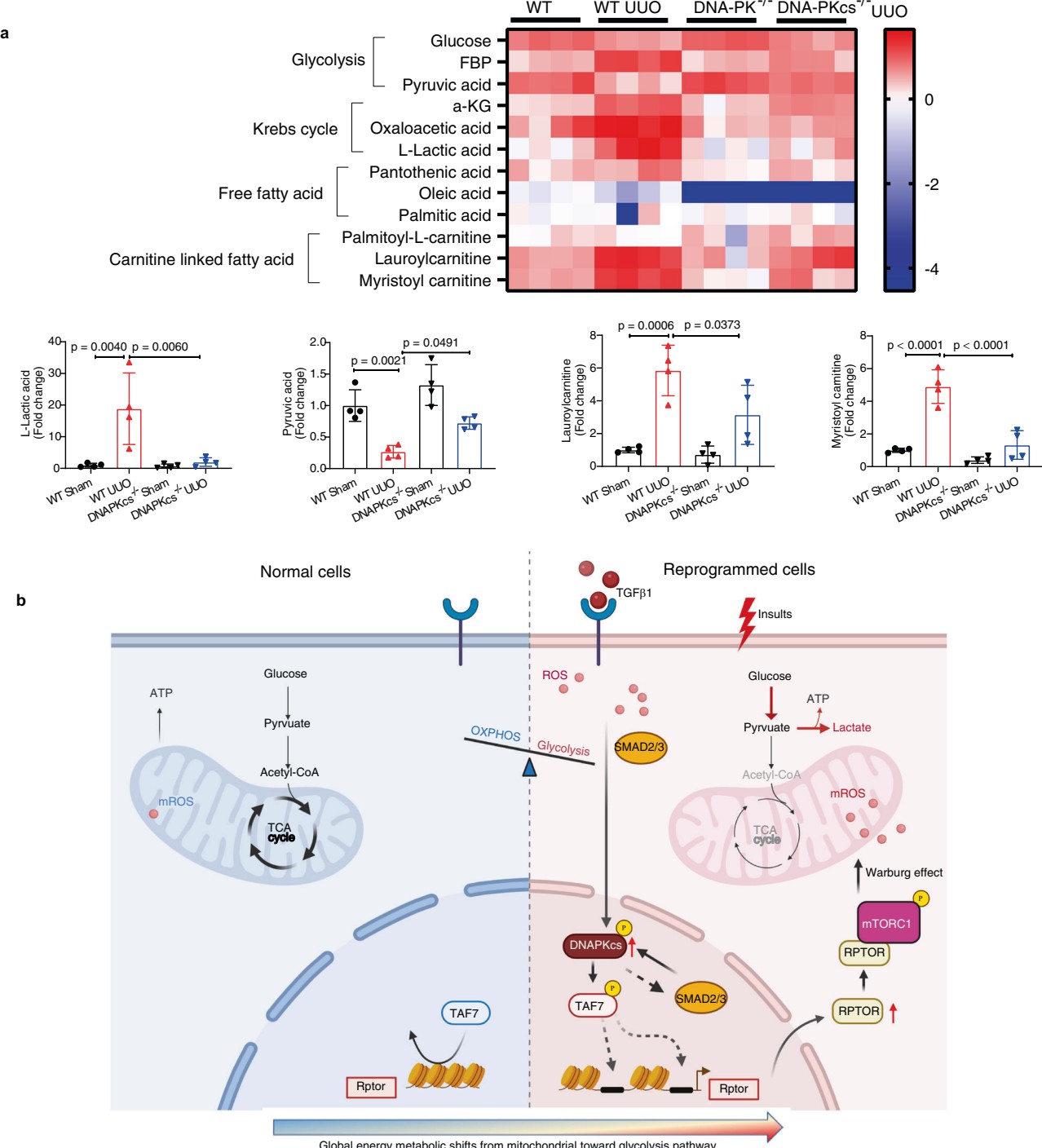

**Fig. 8 | Deletion of DNA-PK corrects metabolic reprogramming.**
**a** Metabolomics analysis of kidney tissues from each group as indicated. Heatmap image showing relative levels of metabolites in the glycolysis pathway, fatty acid metabolism and Krebs cycle in kidneys from each group ($n = 4$). Bars represent statistical analysis of representative metabolites in kidneys of each group (mean ± SD, $n = 4$ mice of each group). One-way ANOVA followed by Tukey's multiple comparisons test was used to determine the *p*-values. **b** Working model (the template was created with BioRender.com) illustrating in which DNA-PKcs mediates activation of Raptor/mTORC1 signaling through phosphorylation of TAF7 and promotes metabolic reprogramming in injured epithelial cells and myofibroblasts.

highlighted that both renal tubular cells and fibroblasts alter metabolic phenotypes in response to CKD[9,10,13]. The shift from fatty acid oxidative pathways and oxidative phosphorylation to glycolysis is the primary characteristic of metabolic reprogramming[9,10]. As mTOR plays a central role in maintaining metabolic homeostasis, inhibition of mTOR displays side effects[55,56]. Moreover, long-term inhibition of TGF-β signaling is also associated with unacceptable adverse effects[57]. In this

study, we found that the activity of DNA-PKcs was almost undetectable in normal kidneys, in contrast to its robust enhancement under CKD. Thus, inhibition of DNA-PKcs corrects metabolic reprogramming in fibrotic kidneys.

In summary, we found that DNA-PKcs mediates the activation of RAPTOR/mTORC1 signaling through phosphorylation of TAF7 and corrects metabolic reprogramming in injured epithelial cells and

myofibroblasts (Fig. 8b). DNA-PKcs may serve as a potential target for treating chronic kidney disease.

## Methods

### Ethical approval

All animal procedures were approved by the Institutional Animal Care and Use Committee of Nanjing Medical University. The protocol concerning the use of human kidney biopsy samples in this study was approved by the Committee on Research Ethics of Children's Hospital of Nanjing Medical University. The written, informed consent to participate was obtained from all study participants (or their parents/legal guardians). All the human study participants agreed to participation for free.

### Human kidney biopsy sample collection and immunostaining

In this study, injured kidney samples obtained from patients with renal fibrosis (clinical parameters of the patients are listed in Supplementary Table 1) and healthy kidney samples used for immunostaining studies were obtained from the Children's Hospital of Nanjing Medical University, China. Healthy control samples were nondiseased portions of tissue from renal cell carcinoma patients who had undergone surgery to remove tumor tissues.

### DNA-PKcs[−/−] mice, Cas9 mice and animal models

DNA-PKcs knockout (DNA-PKcs[−/−]) mice on a BALB/c background were generated and purchased from GemPharmatech (Nanjing, China). Briefly, a 561 bp sequence of *prkdc* (the DNA-PKcs-encoded gene) between exon 6 and exon 7 was deleted using the CRISPR/Cas9 method. DNA-PKcs[−/−] mice and wild-type (WT) littermates were bred in the Laboratory Animal Center of Nanjing Medical University (Nanjing, China) and genotyped using PCR (primers are listed in Supplementary Table 2). Mice were maintained in a standard specific pathogen-free (SPF) animal room free access to food and water with a light/dark cycle of 12 h with room temperature at $21 \pm 2\,°C$ and humidity between 45 and 65%. To construct proximal renal tubular epithelial cells with specific Cas9 transgenic mice, Rosa26-LSL-Cas9 knockin mice (purchased from GemPharmatech, C57BL/6 J background), as described in a previous study[58], were bred with Kap-Cre mice (purchased from Jackson). Kap[+]cas9[+] mice were genotyped using PCR (primers are listed in Supplementary Table 2). To generate renal tubular specific knockout of DNA-PKcs, Male Kap[+]Cas9[+] mice (6–8 weeks old) were given subcapsular injection of adeno-associated virus (AAV8) containing guide RNA (gRNA) for DNA-PKcs ($5 \times 10^{11}$ genome copies per mouse according to a previous study[59]). Kap[+]Cas9[+] mice treated with AAV8 carrying gRNA empty vectors were used as controls. AAV8 carrying gRNA (U6 promoter) for mouse DNA-PKcs (listed in Supplementary Table 2) and empty vectors were purchased from WZ Biosciences Inc (Jinan, China). Mouse renal fibrosis models were generated as previously reported[60]. Briefly, mice were anesthetized using isoflurane (3% for induction and 1.5% for maintenance). For the unilateral ureteral obstruction (UUO) model, DNA-PKcs[−/−] and WT male (male, 6–8 weeks old) were divided into four groups: the UUO and sham control groups of DNA-PKcs[−/−] and WT mice. For UUO surgery, a flank incision was made on the left side of the abdomen, and the left ureter was tied off using two 4.0 surgical silk ties below the renal pelvis. For the sham groups, the surgery was performed as in the UUO group but without ureteral ligation. For unilateral ischemia–reperfusion surgery (UIR), DNA-PKcs[−/−] and WT mice (male, 6–8 weeks old) were divided into four groups: the UIR and sham control groups of DNA-PKcs[−/−] and WT mice. The renal pedicle of the left kidney was clamped using a nontraumatic microaneurysm clamp (RWD Life Science, Shenzhen, China) for 30 min, and the color change of the kidney confirmed successful ischemia–reperfusion. Body temperatures of the mice were maintained at 36.0–38 °C throughout the surgery using a heating instrument.

To examine the effects of a DNA-PK inhibitor (NU7441) in the UUO and UIR mouse models, 6- to 8-week-old male C57BL/6 J mice were purchased from GemPharmatech (Nanjing, China). For the UUO model, one day after surgery, mice oral gavage with or without 40 mg/kg/d NU-7441 in 10% PEG400 in saline until euthanasia once daily. For the UIR model, three days after surgery, mice were dosed once daily by oral gavage with or without 40 mg/kg/d NU-7441 until euthanasia. Mice were euthanized on days 3, 7, or 14 after UUO and on day 21 after UIR. Kidney samples were collected and stored at −80 °C for further analysis. Animal procedures were approved by the Institutional Animal Care and Use Committee of Nanjing Medical University.

### Cell culture and treatment

Human renal tubular epithelial cells (HK-2), mouse renal proximal tubular cells (mPTCs), normal rat kidney interstitial fibroblasts (NRK-49F) and Human Embryonic Kidney 293 (HEK293T) were obtained from American Type Culture Collection (The catalog number of all cell lines are listed in Supplementary Table 3). HK-2 and mPTCs were cultured in DMEM/F-12 medium (Gibco, 319-075-CL), NRK-49F and HEK293T cells were cultured in DMEM. All media were supplemented with 10% fetal bovine serum (FBS, GIBCO), penicillin (100 U/mL) and streptomycin (100 μg/mL) and maintained at 37 °C and 5% $CO_2$ in a humidified incubator. DNA-PKcs or TAF7 knockout cells were constructed using CRISPR/Cas9 methods. Briefly, the sgRNAs targeting DNA-PKcs, TAF7, or KU80 were cloned into pSpCas9 (BB)−2A-Puro (PX459) v2.0, which was a gift from Feng Zhang (Addgene plasmid # 62988) as described in a previous study[61]. Sequences are listed in Supplementary Table 2. The sequenced CRISPR/Cas9 plasmids were transfected into cells using PolyJet™ DNA transfection reagent (SignaGen, SL100688), and puromycin (2 μg mL) was used to select positive cells prior to clonal expansion. Cells transfected with PX459 were used as a negative control. For DNA-PKcs or TAF7 overexpression in cells, human DNA-PKcs plasmids were obtained from Addgene (Addgene plasmid # 83317), and FLAG-tagged human TAF7 plasmids were constructed in this study. All plasmids were transfected into cells using PolyJet. To knock down RPTOR expression in HK2 cells, short interfering RNA (siRNA) of RPTOR was transfected with Lipofectamine2000 (Thermofisher). Cells were treated with human recombinant TGF-β1 (100-B-010-CF, R&D Systems) in serum-free medium for 24 h or the indicated time.

### Primary renal epithelial cell isolation

Primary kidney epithelial cells were isolated from DNA-PKcs[−/−] mice or WT littermates as previously reported[13]. Briefly, male mice (3 to 5 weeks old) were euthanized, and the kidneys were immediately collected and placed in cold HBSS with 1% penicillin and streptomycin. Then, the kidneys were minced into pieces of approximately 1 mm³ and digested in 5 ml HBSS containing 2 mg/mL collagenase I for 30 min at 37 °C. The supernatants were strained through a 100 μm nylon mesh and then centrifuged for 10 min at 900 g and 4 °C. The primary cells were cultured with RPMI 1640 supplemented with 10% FBS, 20 ng/mL EGF (Sigma, St. Louis, MO) and 100 units/mL penicillin and 100 μg/mL streptomycin maintained at 37 °C and 5% $CO_2$ in a humidified incubator. Cells were used after 7 days of culture.

### Western blot analysis

Total proteins from kidneys or cultured cells were extracted in RIPA lysis buffer (Beyotime, A0181, China) containing 1× protease inhibitor cocktail (Roche, 04693132001) for 30 min on ice. The lysates were collected after centrifugation at 13,800 g for 15 min at 4 °C. Protein concentrations were determined using the BCA Protein Assay Kit (Beyotime, P0012), and equal total protein (30–50 μg) of each sample was analyzed using a standard western blot assay. Briefly, protein samples were separated on 4–20% polyacrylamide separating gels, and separated proteins were electroblotted onto PVDF membrane. The

membranes were blocked in 5% nonfat milk for 1 h at RT (room temperature) followed by incubation with specific primary antibodies (1:1000 dilution) overnight at 4 °C. The next day, membranes were incubated with peroxidase-conjugated goat anti-rabbit (Beyotime; A0208, 1:1000 dilution) or anti-mouse (Beyotime; A0216, 1:1000 dilution) at RT for 1 h. The bands were visualized using an enhanced chemiluminescence detection system (Bio–Rad, Hercules, CA, USA). Protein band densitometry analysis was performed using ImageJ (Wayne Rasband National Institutes of Health, USA), and relative protein expression levels were normalized to GAPDH. Phos-tag SDS-PAGE was performed as previously reported[62,63]. Briefly, 100 μM Phosbind (Apexbio) was added to 8% (w/v) polyacrylamide separating gels before polymerization, according to the manufacturer's instruction, to separate phosphorylated isoforms of TAF7 (all the antibodies, reagents, and their dilution are listed in Supplementary Table 3).

### Quantitative real-time PCR (qRT-PCR)

Total RNA was extracted from cultured cells or kidney tissues using TRIzol reagent (TAKARA, Dalian, China; 9108). Subsequently, total RNA (1 μg) was reverse-transcribed to cDNA using a reverse transcriptase M-MLV kit (TAKARA, 2641A). Quantitative real-time PCR amplification (qRT–PCR) was performed using SYBR Green master mix (Vazyme, Nanjing, China; q111-02/03) in a 96-well QuantStudio 3 Real-time PCR System (Applied Biosystems, Foster City, CA, USA). $2^{-\Delta\Delta Ct}$ method was used to calculate relative expression levels of messenger RNA (mRNA) as previously study described[64]. Relative expression levels of the target genes were normalized to glyceraldehyde 3-phosphate dehydrogenase (GAPDH) levels. Primer sequences (listed in Supplementary Table 2) were designed and synthesized by Tsingke (Nanjing, China).

### Luciferase reporter assay

The promoter sequences of human DNA-PKcs and Raptor were amplified and cloned into the pGL3 basic vector (Promega Corporation) using a ClonExpress Ultra One Step Cloning Kit (Vazyme, China, C115-01). The primer sequences are listed in Supplementary Table 2. Raptor luciferase reporter plasmids, PRL (Renilla luciferase) and two mutants at position 213 of TAF7 (TAF7-WT, TAF7-S213D, and TAF7-S213A, respectively) plasmids were co-transfected into HK-2 cells with PolyJet™ DNA transfection reagent. After transfection for 24 h, cells were treated with or without TGF-β1 (5 ng/ml) for another 24 h. The samples were subsequently harvested, and luciferase activity was measured using a Dual-Luciferase Reporter Assay System (Promega Corporation). Relative luciferase activity was normalized to renilla luciferase activity of the sample. To analyze transcriptional activation of DNA-PKcs by SMAD2, DNA-PKcs promoter plasmids and human SMAD2 plasmids (GeneCopoeia) were co-transfected into HEK293T cells for Dual-Luciferase Reporter Assay analysis.

### ChIP assay

ChIP assay was performed by using a SimpleChIP® Enzymatic Chromatin IP Kit (9003, CST) according to the manufacturer's instruction. Briefly, HEK293T cells were transfected with FLAG-tagged TAF7 for 24 h and cross-linked with 37% formaldehyde solution for 10 min at RT. Then, the chromatin extract was digested with micrococcal nuclease and incubated with anti-FLAG ChIP antibody (CST) overnight at 4 °C. The sequence containing the TAF7-binding site in the promoter of human RPTOR was amplified by PCR using ChIP products as template, and the relative signal was normalized to the level of input (total chromatin extract) by using the same primers (Supplementary Table 2) targeting RPTOR promoter. Histone H3 antibody was used as a positive control and rabbit IgG isotype as a negative control.

### Immunoprecipitation

The immunoprecipitation assay was performed as previously described[65]. Briefly, after human DNA-PKcs plasmids and TAF7-FLAG

plasmids were cotransfected into HK-2 cells using PolyJet™ DNA transfection reagent for 36 h, the cell lysates were prepared and incubated with the indicated antibodies overnight at 4 °C with gentle rotation, followed by incubation with protein A&G beads (Santa Cruz) for 2 h with gentle rotation. The beads were washed with cold 1× lysis buffer for three times, and the immunoprecipitation complexes were collected by centrifugation and subjected to Western blot analysis.

### Measurement of ECAR using an XF96 Flux Analyzer

The extracellular acidification rate (ECAR) of the cultured cells was analyzed using a Seahorse XF96 Extracellular Flux Analyzer (Seahorse Bioscience, Copenhagen, Denmark). Briefly, cells were seeded into XF96 Cell Culture Microplates (Seahorse Bioscience) at a density of 5000 cells per well. Twenty-four hours later, the cells were treated with or without NU7441 (0.1 μM) and stimulated with TGF-β1 (5 ng/ml) for 24 h in serum-free medium. Real-time ECAR was analyzed as follows: basal ECAR was recorded for 16 min, followed by sequential injections with glucose (10 mM), oligomycin (5 μg/ml), and 2-DG (50 mM).

### Extracellular lactate measurement

Briefly, cells were seeded into 6-well plates until 70% confluence, treated with or without NU7441 (0.1 μM) and stimulated with TGF-β1 (5 ng/ml) for 24 h in serum-free medium. Extracellular levels of lactate were measured using a lactate assay kit (E-BC-K044-M, Elabscience, China) according to the manufacturer's instructions.

### Histological examination of kidney tissue

Kidney tissues were fixed in 4% paraformaldehyde (PFA) for 48 h at RT. Kidney sections (4 μm thick) were prepared for periodic acid-Schiff (PAS) staining and Masson's trichrome staining, kidney sections (6 μm thick) for Sirius red staining. PAS, Masson, and Sirius red staining were performed following standard protocols as previously described[8]. The kidney injury index was evaluated by calculating the percentage of damaged renal tubules that exhibited cell lysis or loss of the brush border via PAS staining. The pathological damage of kidneys was scored from 0 to 4: 0, normal; 1, <25% damaged renal tubules; 2, 25–50% damaged renal tubules; 3, 50–75% damaged renal tubules; 4, >75% of damaged renal tubules[66]. The PAS-stained images of each sample were captured using an Olympus BX51 microscope (Olympus, Center Valley, PA), and three random visual fields of each sample were selected and quantitated. For analysis of the fibrotic area, Masson staining images were captured using an Olympus BX51 microscope, and quantitative evaluation was performed using ImageJ (Wayne Rasband National Institutes of Health, USA). The collagen-stained area was calculated as a percentage of the total area, and at least five randomly chosen kidney cortex images (×400) were examined from each mouse.

### Immunohistochemistry (IHC) and immunofluorescence (IF) staining

IHC staining was performed on kidney paraffin sections as previously described[67]. Briefly, 4 μm paraffin-embedded kidney sections on slides were deparaffinized, followed by antigen retrieval and rehydration. Then, the sections were blocked in 3% $H_2O_2$ for 15 min, washed with TBST buffer, and blocked in 10% normal goat serum for 1 h before incubation with the indicated primary rabbit antibody (listed in Supplementary Table 3) overnight at 4 °C. After washing three times with TBST buffer, the sections were incubated with SignalStain® Boost IHC Detection Reagent (CST, 8114) for 1 h. Finally, the peroxidase conjugates were stained using a DAB kit (ZLI-9018, Zsbio, China), and images were captured with an Olympus BX51 microscope. The relative positive areas of IHC images were analyzed using ImageJ. For immunofluorescence staining performed on paraffin-embedded kidney sections, the deparaffinization, antigen retrieval and rehydration steps were the same as for IHC staining. The sections were incubated with the indicated primary rabbit antibody overnight at 4 °C. The next day,

goat anti-rabbit IgG (H + L) cross-adsorbed secondary antibody, Alexa Fluor 488 (Thermo Fisher Scientific, A-11008), was used as a secondary antibody according to the manufacturer's instructions. Nuclei were counterstained with DAPI (Beyotime, P0131). Images were obtained using an LSM710 confocal microscope (CarlZeiss, Germany). For IF staining, cells were seeded onto polylysine-coated glasses. After treatment, the cells were fixed in 4% paraformaldehyde for 10 min, permeabilized and blocked with 0.1% Triton™ X-100 dissolved in 1% BSA for 1 h, and the remaining steps were the same as IF staining of paraffin-embedded kidney sections. Quantification was performed using ImageJ.

## RNA sequencing analysis

RNA samples were collected from TAF7 knockout (TAF7$^{-/-}$) or control mPTCs treated with or without TGFβ1 (5 ng/ml) for 24 h. RNA isolation, library construction, and sequencing were performed by BGI (Beijing Genomic Institution, www.genomics.org.cn, BGI) using a BGISEQ-500 RNA-seq platform. Through the BGI bioinformatics platform, differentially expressed genes among the four groups, NC (negative control), NC + TGFβ1, TAF7$^{-/-}$ and TAF7$^{-/-}$ + TGFβ1, were analyzed using several bioinformatics methods, including gene set enrichment analysis (GSEA), KEGG pathway enrichment analysis and heatmap analysis.

## Phosphoproteomics analysis

Phosphoproteomics analysis was performed by PTM Biolabs (PTM Biolabs, Hangzhou, China). Briefly, kidney protein samples from DNA-PKcs$^{-/-}$ and WT mice were prepared and ground in liquid nitrogen into powder and then prepared in protein solution for trypsin digestion. A bio-material-based method was used for phosphopeptide enrichment. Briefly, peptide mixtures were first incubated with IMAC microspheres suspension in loading buffer (50% acetonitrile/6% trifluoroacetic acid) with shaking gently. The IMAC microspheres with enriched phosphopeptides were collected by centrifugation, and the supernatant was discarded. Next, the IMAC microspheres were sequentially washed with 50% acetonitrile/6% trifluoroacetic acid and 30% acetonitrile / 0.1% trifluoroacetic acid to remove nonspecifically adsorbed peptides. At last, elution buffer containing 10% NH4OH was added to elute the enriched phosphopeptides from the IMAC microspheres with vibration. The enriched phosphopeptides supernatant was collected and lyophilized for LC-MS/MS analysis. The phosphopeptides were loaded to a nitrogen solubility index (NSI) source followed by tandem mass spectrometry (MS/MS) in Q Exactive (Thermo Fisher Scientific, San Jose, CA, USA) coupled to an online ultra-performance liquid chromatography (UPLC) system. The MS/MS data were analyzed using the MaxQuant search engine (v.1.5.2.8), carbamidomethylated cysteine residues (C, + 57.0340 Da) as a fixed modification, Oxidation (M, + 15.9949 Da), phosphorylated serine (S), threonine (T) and tyrosine (Y) (+79.9663 Da) and Acetyl (Protein N-term, +42.011 Da) as variable modifications. FDR was adjusted to <1%, and the minimum score for modified peptides was set >40. Bioinformatics methods, including GO Annotation, Motif Analysis, Functional Enrichment, Enrichment-based Clustering and Protein–protein Interaction Network, were used to analyze differentially expressed modified proteins between DNA-PKcs$^{-/-}$ and WT mice.

## In vitro kinase assay and mass spectrometry analysis

To examine whether TAF7 could be phosphorylated by DNA-PKcs directly, an in vitro kinase assay was performed. First, human DNA-PKcs was purified from TGFβ1 treated HK2 cells by immunoprecipitation with a DNA-PKcs antibody (Abcam) and IgG was used as a negative control. The purified proteins and human GST-TAF7 (purchased from Proteintech) were incubated in kinase assay buffer (50 mM Tris-HCl, pH 7.5, 150 mM NaCl, 10 mM MgCl2, and 1 mM MnCl2) supplemented with 200 μM ATP in 50 μl reactions for 30 min at 30 °C. Then, phosphorylation of TAF7 was analyzed by mass spectrometry (MS), which

was performed by BIOTREE (Shanghai, China). MS analysis was performed as described in a previous study[68]. Proteome Discoverer (PD) software and the built-in Sequest HT search engine were used to process raw MS files. MS spectra lists were searched against their UniProt FASTA databases (Homo sapiens-9606-2021-8. fasta), carbamidomethylated cysteine residues (C, + 57.0340 Da) as a fixed modification, Oxidation (M, + 15.9949 Da), phosphorylated serine (S), threonine (T) and tyrosine (Y) (+79.9663 Da) and Acetyl (Protein N-term, +42.011 Da) as variable modifications of peptides.

## Metabolomics analysis

Metabolomics analysis of kidney tissues was performed by BioNovogene (Suzhou, China). Briefly, kidney tissues were prepared for LC−MS detection following a standard method by BioNovogene. Chromatographic separation was performed in a thermo vanquish system equipped with an ACQUITY UPLC® HSS T3. The MS experiments were executed on a Thermo Q Exactive Focus mass spectrometer with spray voltages of 3.5 kV and −2.5 kV in positive and negative modes, respectively. Differentially abundant metabolites were analyzed using bioinformatics methods, and all procedures were performed by Bio-Novogene (Suzhou, China).

## Statistical analysis

Data are shown as the mean ± standard deviation (SD) and were analyzed using GraphPad Prism 9.0. Statistical significance between two groups was determined using unpaired Student's $t$ test. When more than two groups were compared, one-way or two-way ANOVA followed by Tukey's or Šídák's multiple comparison test was used to analyze differences between two groups of interest. A $P$ value less than 0.05 was considered statistically significant.

## Reporting summary

Further information on research design is available in the Nature Portfolio Reporting Summary linked to this article.

## Data availability

Publicly available data used in this paper were obtained from Nephroseq[69], a single-cell sequencing database[35] (http://humphreyslab.com/SingleCell/) and UniProt FASTA database (https://www.uniprot.org/). For the data generated in this study, the RNA sequencing data has been submitted to National Center for Biotechnology Information (NCBI) Sequence Read Archive (SRA) database with the identifier PRJNA794204. Phosphoproteomic data of DNA-PKcs knockout and control kidney tissues has been submitted to iProX with the identifier PXD030789. Mass spectrometry analysis of in vitro kinase assay has also been submitted to iProX with the identifier PXD036930. Metabolomics data of kidney tissues have been submitted to metabolights[70] (www.ebi.ac.uk/metabolights/MTBLS5971). All other data generated or analyzed during this study are included in this published article (and its supplementary information files). Source data are provided with this paper.

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

## Acknowledgements

This study was supported by the National Key Research and Development Program (2022YFC2705100, 2019YFA0802700, 2019YFA0802702) to A.Z., the National Natural Science Foundation of China (81530023 to A.Z., 82090022 to A.Z., 81830020 to A.Z., 82070701, 82270773, 81873599 to Z.J., 81700642 to Y.Y., 81974084 to S.H., 82170754 to Y.Z.), and Nanjing Health Technology Development Program (ZKX22050 to Y.Y.). We thank Dr. Fangfang Cai for providing kindly helps in protein docking.

## Author contributions

Y.Y., S.L., P.W., and J.O. performed the experiments and prepared the figures. S.H., Z.J., A.Z., and Y.Y. designed the experiments, analyzed the data, and wrote the main text of the manuscript. N.Z. and Y.Z. offered the assistance with the manuscript preparation, and all authors read and approved the final manuscript.

## Competing interests

The authors declare no competing interests.
