## [Peer Review File · Nature Communications]

DNA-PKcs drives CKD progression by activating TAF7/RAPTOR/mTORC1 signaling-mediated metabolic reprogrammingREVIEWER COMMENTS

Reviewer #1 (Remarks to the Author):

Yang and co-workers identify a novel pro-fibrotic signaling axis in the kidney, in which the DNA-dependent protein kinase (DNA-PK) phosphorylates the transcriptional activator TAF7, which induces Raptor expression, which causes mTORC1 activation, which determines metabolic reprogramming of kidney cells which ultimately contributes to progression of fibrosis in the kidney.

This is a substantial body of work, methods are state-of-the-art and identification of each downstream target is plausible. A few conceptual questions and controls require attention.

Specific comments:

Ultimately, this study comes down to metabolic reprogramming as a driver of fibrosis and identification of a novel pathway, which causes such metabolic reprogramming. Hence, Figures 7e-g are really the most critical part of this study. However, observed changes in gene expression appear relatively minor. This should be further substantiated.

The rationale of why DNA-PK was studied to begin with is unclear and the increase of DNA-PK in human kidneys is rather subtle. Is DNA-PK even upregulated within fibroblasts? I recommend separate analysis of DNA-PK in tubular epithelial cells and fibroblasts.

Additional analysis of human samples (e.g. p-DNA-PK, TAF7 immunostaining in renal pathologies) would further strengthen the translational relevance of the findings.

The issue of impaired leukemic cells in DNA-PK needs to be further clarified. Detailed characterization of the mononuclear infiltrates within the different mouse cohorts would be a good start.

Complete absence of phenotypic changes in response to TGFbeta in absence of DNA-PK is puzzling.

Considering the substantial body of literature which linked TGFbeta-induced fibroblast activation and EMT to Smad-signaling – How does absence of DNA-PK completely block this effect?

A major novelty of this study is a novel function of DNA-PK. This concept would be substantially strengthened if the authors could re-create metabolic reprogramming through the DNA-PK-TAF7-Raptor-mTORC1 axis in an unstressed DSB-free system.

Additional comments:

As control experiments, DNAPKcs knockout mice should be treated with NU7441 to confirm specific action on DNA-PK.

To confirm that DNA-PK-mediated phosphorylation of TAF7 promotes mTORC1 is the proposed mechanism, in vitro studies in tubular epithelial cells and fibroblasts should also include control experiments with NU7441 exposure.

Reviewer #2 (Remarks to the Author):

In this study, Yunwen Yang et al found that DNA-PK was upregulated in the fibrotic kidneys from patients and animal models. Blockade of DNA-PK with genetic method or pharmacologic inhibitor attenuated fibroblast activation and kidney fibrosis. In addition, the authors also proved that DNA-PK

promoted kidney fibrosis through stimulating TAF7 mediated mTORC1 activation and subsequently metabolic reprogramming. There are several concerns should be addressed.

- 1) Besides FN, The association of DNA-PK expression level with fibrotic area in kidneys of the CKD patients should be presented.
- 2) How dose TGFb1 stimulate DNA-PK upregulation or activation in tubular cells and fibroblasts?
- 3) The expression level and location of exogenous DNA-PK after plasmid injection in the mouse kidneys should be presented.
- 4) The detail mechanisms for TAF7 regulating RAPTOR expression are not clear.
- 5) Although TAF7 could regulate RAPTOR expression, whether RAPTOR induction mediates DNA-PK/TAF7 stimulated mTORC1 activation and the metabolic reprogramming should be further determined.
- 6) Whether DNA-PK affects Smad signaling should be tested.
- 7) All of the genetic models are global knockouts, the tubular cell specific or fibroblast specific gene deletion model should be employed to decipher the role for DNA-PK or TAF7 in kidney fibrosis.
- 8) In Fig 1g, there are two bands for the WB of DNA-PK, which one is DNA-PK?
- 9) Fig 2 h,i,j and k are not labeled correctly.
- 10) In Fig 3, The western blot image for TGFb1 stimulated DNA-PK expression and phosphorylation in cultured cells should be presented.

Reviewer #3 (Remarks to the Author):

The authors begin by showing that DNA-PK expression and the presence of its S2056 phosphorylated form is markedly increased in the kidneys of CKD patients and CKD-model mice. They go on to show that DNA-PK knockout or treatment with the DNA-PKcs inhibitor, NU7441, attenuates CKD in mice, and initiate in the next step in vitro experiments to uncover the underlying molecular mechanisms aiming at novel therapies for the disease.

In the in vitro part of their work the authors show that DNA-PK deficiency preserves epithelial cell phenotype and inhibits fibroblast activation caused by TGFβ1. This is relevant because in injured kidneys, interstitial fibrosis involves TGFβ1 induction, that causes epithelial cell dedifferentiation and myofibroblast activation. They present evidence that a putative DNA-PK substrate, TAF7, promotes mTORC1 activation by upregulating the expression of RAPTOR, which in-turn promotes metabolic reprogramming in injured epithelial cells and myofibroblasts. They demonstrate that inhibition of DNA-PK corrects metabolic reprogramming by regulating TAF7/mTORC1 signaling in CKD, offering new ways for treating the disease.

The paper presents interesting and novel results that uncover unknown facets of DNA-PK function. The possibility that DNA-PK inhibitors may help to treat the disease is also exciting and worth pursuing. The paper summarizes a great deal of work that appears well conducted, and is overall very well written. The results shown justify the conclusions drawn, as the reported effects are uniformly large.

Specific comments:

1. From the perspective of this reviewer one missing aspect in the work concerns the mode of DNA-PK

activation. As the authors point out, the typical activator, DSBs, appear not to be involved in the observed effects. It will certainly be unrealistic to ask the authors to delve in this direction as part of the present paper, however, one aspect that could certainly be added without much effort and which will significantly enhance the message of the paper, is to test whether KU is actually contributing to the observed effects. The authors, following the dogma of the field, assume that this is the case, i.e. that KU is required. However, because γ H2AX foci are not seen at any stage of the disease or in the extensive set of the reported in vitro experiments, it will be tantalizing to investigate whether DNA-PKcs actually functions in this setting independently of KU. This could be readily investigated in a set of knockdown experiments, where KU components of knocked down and selected endpoints investigated.

2. The connection between DNA-PKcs and the activation of TGF β 1 could be analyzed in greater detail. DNA-PKcs has been shown to interact with internalized growth factors and it will be interesting to see whether this is also the case with TGF β 1.

3. The broad effects of systemic DNA-PKcs ablation in the mouse leave room for indirect contributions to the observed endpoints. The authors address this point in the Discussion. Have the authors also considered organ specific ablation of the protein – for example in the kidney. Such a mouse model will provide a perfect tool for experiments designed to develop therapies for CKD. The authors may wish to discuss this aspect.

4. Thinking about therapeutic application of the information in the paper: can the authors speculate where they hope to get more mileage: Inhibition of DNA-PK, inhibition of TGF β 1, or inhibition of mTor?

Reviewer #4 (Remarks to the Author):

The manuscript by Jia et al. describes observations that DNA-PK is upstream of CKD via a TAF7/RAPTOR/mTOR signalling process. The authors claim that TAF7 is a phosphorylation substrate of DNA-PK. The evidence for this is not sufficiently investigated. First, the authors performed phosphoproteomics analysis in DNA-PK deficient mice model compared to control. The technical details for phosphopeptide and phosphorylation site identification are not provided. Typically, phosphoproteins and the identification of phosphorylation sites require phosphoprotein/peptide enrichment because of low-medium abundance. It is not clear how and which phosphorylation sites in the kidney tissues were identified. It is claimed that among these proteins TAF7 phosphorylation was decreased. One can only guess that the authors refer to phosphorylation site 213, from figure 5. Furthermore, any up-and downregulation of TAF7 is then claimed by western blot analysis of the TAF7 protein and not the phosphorylated form by saying that the phosphorylated TAF7 is 55kDa instead of 40kDa referring to literature. However, from the cited literature this is not clear at all. A direct analysis of the phosphorylation site of TAF7 is necessary. The authors need to identify and quantify the direct phosphorylation site by mass spectrometry. It is also not validated by the data that DNA-PK phosphorylates TAF7, there is only indirect evidence. Here more in vivo and in vitro studies are necessary.

Response to reviewers:

Reviewer #1 (Remarks to the Author):

Yang and co-workers identify a novel pro-fibrotic signaling axis in the kidney, in which the DNA-dependent protein kinase (DNA-PK) phosphorylates the transcriptional activator TAF7, which induces Raptor expression, which causes mTORC1 activation, which determines metabolic reprogramming of kidney cells which ultimately contributes to progression of fibrosis in the kidney. This is a substantial body of work, methods are state-of-the-art and identification of each downstream target is plausible. A few conceptual questions and controls require attention.

Specific comments:

1, Ultimately, this study comes down to metabolic reprogramming as a driver of fibrosis and identification of a novel pathway, which causes such metabolic reprogramming. Hence, Figures 7e-g are really the most critical part of this study. However, observed changes in gene expression appear relatively minor. This should be further substantiated.

Response: Thank you very much for your valuable suggestion. The protein levels of key enzymes involved in fatty acid oxidation including CPT1A and ACOX1 were also analyzed by Western blot and the results (**Figure 7b & c**) were provided in our revised manuscript. Accordingly, the data about the mRNA levels of key enzymes involved in fatty acid oxidation were moved to **Supplementary Fig. 8**.

2. The rationale of why DNA-PK was studied to begin with is unclear and the increase of DNA-PK in human kidneys is rather subtle. Is DNA-PK even upregulated within fibroblasts? I recommend separate analysis of DNA-PK in tubular epithelial cells and fibroblasts.

Response: As one of the phosphoinositide 3-kinase (PI3K)-related kinases, DNA-PKcs has been found to regulate mTOR activation, although the underlying mechanism needs further research ¹. These attract our interest to examine the possible role of DNA-PK in CKD. We have added these in the Introduction section and highlighted them in red. Furthermore, we have analyzed the activity of DNA-PKcs in tubular epithelial cells and fibroblasts separately according your suggestion. The results showed that p-DNA-PKcs were upregulated in tubular epithelial cells and myofibroblasts of human CKD kidney tissues (**Fig. 1d & 1e**).

3. Additional analysis of human samples (e.g. p-DNA-PK, TAF7 immunostaining in renal pathologies) would further strengthen the translational relevance of the findings.

Response: Thank you very much for your valuable suggestion. Co-staining of p-DNA-PKcs with TAF7 with immunofluorescence was performed, and the results revealed a positive association between the protein levels of p-DNA-PKcs and TAF7 in human CKD kidney tissues (**Fig. 5c** in the revised manuscript).

4. The issue of impaired leukemic cells in DNA-PK needs to be further clarified. Detailed characterization of the mononuclear infiltrates within the different mouse cohorts would be a good start.

Response: Mutations in DNA-PKcs arrest the development of T and B lymphocyte, due to the deficiency of V (D) J recombination ^{2,3}. However, growth retardation or high frequency of T cell lymphoma development did not appear in DNA-PKcs knockout mice ³. The content about impaired leukemic cells in DNA-PKcs knockout mice was added into the Introduction section and highlighted in red in revised manuscript. To

exclude the potential effect of lymphocyte deficiency, we constructed proximal renal tubular epithelial cells with specific DNA-PKcs knockout *in vivo* by using CRISPR/cas9 knockin mice. Our results revealed that renal tubular specific deletion of DNA-PKcs also attenuated the progression of renal interstitial fibrosis in UUO (**Fig. 2g & h**).

5. Complete absence of phenotypic changes in response to TGFbeta in absence of DNA-PK is puzzling. Considering the substantial body of literature which linked TGFbeta-induced fibroblast activation and EMT to Smad-signaling – How does absence of DNA-PK completely block this effect?

Response: Thanks for your comments. To answer these questions, we also examined the possible mechanism of TGFβ1-induced DNA-PKcs expression. Firstly, we found multiple possible binding sites of SMAD2 in the promoter region of DNA-PKcs, through JASPAR transcription factor binding profiles analysis⁴. Secondly, luciferase assay results showed that overexpression of SMAD2 increased transcriptional activation of human DNA-PKcs (**Supplementary Fig. 5e**), suggesting a transcriptional regulation of SMAD2 on DNA-PKcs. Additionally, the results showed DNA-PKcs knockout greatly inhibited the activation of SMAD2/SMAD3 in UUO mouse model (**Supplementary Fig. 5f**), although the detail mechanism remains unclear. All these results suggest DNA-PKcs-SMAD signaling perhaps form a positive loop under the CKD condition (**Supplementary Fig. 5g**).

6. A major novelty of this study is a novel function of DNA-PK. This concept would be substantially strengthened if the authors could re-create metabolic reprogramming

through the DNA-PK-TAF7-Raptor-mTORC1 axis in an unstressed DSB-free system.

Response: Thank you very much for your valuable suggestion. KU70/80 first recognizes DNA broken ends at DSB, then recruits DNA-PKcs to form DNA-PK that is necessary in the repair of DSB via NHEJ⁵. As we have no unstressed DSB-free system at this time, we constructed KU80 knockout tubular epithelial cells, and the results showed KU80 knockout boosted FN production induced by TGFβ1 in tubular epithelial cells, which was opposite to the results of DNA-PKcs knockout (**Fig. 4m**). Moreover, DNA-PKcs knockout did not aggravate DNA breaks in UO mice and in HK2 cells. These results suggest that DNA-PKcs may mediate the profibrotic response of renal epithelial cells independent of its DSB repair function. For this reason, we revised DNA-PK as DNA-PKcs in the Title of manuscript.

Additional comments:

1. As control experiments, DNAPKcs knockout mice should be treated with NU7441 to confirm specific action on DNA-PK.

Response: Thank you very much for your valuable suggestion. DNA-PKcs knockout mice were also treated with NU7441 to examine the possible off-target of NU7441. Our results showed that NU7441 had no additional anti-fibrosis effects in DNA-PKcs knockout mice (**Fig. 3g & h**). These results confirmed the specific action of NU7441 on DNA-PKcs.

2. To confirm that DNA-PK-mediated phosphorylation of TAF7 promotes mTORC1 is the proposed mechanism, in vitro studies in tubular epithelial cells and fibroblasts should also include control experiments with NU7441 exposure.

Response: The control experiments based on tubular epithelial cells (**Fig. 5h**) and fibroblasts (**Fig. 5i**) under NU7441 exposure were provided in the revised manuscript.

Reviewer #2 (Remarks to the Author):

In this study, Yunwen Yang et al found that DNA-PK was upregulated in the fibrotic kidneys from patients and animal models. Blockade of DNA-PK with genetic method or pharmacologic inhibitor attenuated fibroblast activation and kidney fibrosis. In addition, the authors also proved that DNA-PK promoted kidney fibrosis through stimulating TAF7 mediated mTORC1 activation and subsequently metabolic reprogramming. There are several concerns should be addressed.

1. Besides FN, The association of DNA-PK expression level with fibrotic area in kidneys of the CKD patients should be presented.

Response: Thank you very much for your valuable suggestion. Sirius red staining was also performed to analyze the degree of interstitial fibrosis in CKD patients (**Fig. 1a**). The association of p-DNA-PKcs expression level with fibrotic area in kidneys of the CKD patients was presented (**Fig. 1c**).

2. How dose TGFb1 stimulate DNA-PK upregulation or activation in tubular cells and fibroblasts?

Response: In the revised manuscript, we also examined the possible mechanism of TGFβ1-induced DNA-PKcs expression. Firstly, we found multiple possible binding sites of SMAD2 in the promoter region of DNA-PKcs through JASPAR transcription factor binding profiles analysis ⁴. Secondly, luciferase assay results showed that overexpression of SMAD2 enhanced transcriptional activation of human DNA-PKcs

(**Supplementary Fig. 5e**). Thus, above data suggested a transcriptional regulation of TGF β 1-Smad2 signaling on DNA-PKcs.

3. The expression level and location of exogenous DNA-PK after plasmid injection in the mouse kidneys should be presented.

Response: In this study, human DNA-PKcs overexpression plasmids (Addgene: 83317) were delivered into mouse kidneys through a tail vein high-pressure injection method, as described in several previous studies ⁶. Exogenous DNA-PKcs was analyzed by immunofluorescence stained with a human-specific DNA-PKcs antibody (abcam, ab133516) and the results showed exogenous DNA-PKcs mainly expressed in renal tubular cells (**Supplementary Fig. 3h**).

4. The detail mechanisms for TAF7 regulating RAPTOR expression are not clear.

Response: We performed ChIP assay and the results showed that TAF7 could bind to *Raptor* promoter directly (**Fig. 6e**). Although TAF7 itself is not a core transcription factor, these results suggest TAF7 may collaborate with other transcription factors to express *Raptor*.

5. Although TAF7 could regulate RAPTOR expression, whether RAPTOR induction mediates DNA-PK/TAF7 stimulated mTORC1 activation and the metabolic reprogramming should be further determined.

Response: RPTOR (RAPTOR) is a positive regulator of mTORC1 ^{7,8}. To examine the role of RAPTOR in mediating mTORC1 activation stimulated by DNA-PK/TAF7, RAPTOR expression was knocked down by siRNA in tubular epithelial cells. The results showed knockdown of RAPTOR markedly inhibited the phosphorylation of

mTOR induced by TGF β 1 (**Supplementary Fig. 7h**). This result further confirmed a regulation of RAPTOR on mTORC1 activation, which is known in regulating metabolic reprogramming^{9,10}.

6. Whether DNA-PK affects Smad signaling should be tested.

Response: Thanks for the comment. We analyzed whether DNA-PK affects Smad signaling according to your suggestion, and the results showed DNA-PKcs knockout inhibited the activation of SMAD2/SMAD3 in UUO mouse model (**Supplementary Fig. 5f**).

7. All of the genetic models are global knockouts, the tubular cell specific or fibroblast specific gene deletion model should be employed to decipher the role for DNA-PK or TAF7 in kidney fibrosis.

Response: Thanks for the comment. According to your suggestion, we constructed proximal renal tubular epithelial cells with specific DNA-PKcs knockout *in vivo* by using CRISPR/cas9 knockin mice (**Supplementary Fig. 2e, f**) and Kap-cre⁺ Cas9⁺ mice, as described in the Methods. Three weeks after subcapsular injection of adeno-associated virus (AAV8) containing guide RNA (gRNA) for DNA-PKcs, the proximal renal tubular specific DNA-PKcs knockout mice were generated, subjected to UUO and euthanized 7 days after surgery. The immunofluorescence staining of DNA-PKcs showed a lower DNA-PKcs expression in renal tubular of cas9 mice injected with AAV-gRNA, compared to cas9 mice injected with an AAV without gRNA for DNA-PKcs (**Fig. 2h**). Protein levels of DNA-PKcs and profibrotic markers, including FN and α -SMA, were also significantly reduced in the UUO kidneys of cas9 mice injected with

AAV-sgRNA, as analyzed by western blot (**Fig. 2g**).

8. In Fig 1g, there are two bands for the WB of DNA-PK, which one is DNA-PK?

Response: In this study, we used tissues from DNA-PKcs knockout mice to validate the antibody purchased from Abcam (ab32566). Additionally, Western blot analysis showed two bands of DNA-PKcs over 200 kDa in control group, but two bands disappeared in DNA-PKcs knockout cells (figure b, below) and kidney tissues of DNA-PKcs knockout mice subjected to UUO (figure c, below). Thus, the modification or cleavage of DNA-PKcs might result in two bands over 200 kDa.

Figure legend: a, Western blot analysis of the protein levels of DNA-PKcs in kidney tissues of WT (wild-type) mice (C57BL/6) subjected to UUO (3d, 7d, 14d); b, Western blot analysis of the protein levels of DNA-PKcs in DNA-PKcs knockout HK2 cells; c, Western blot analysis of the protein levels of DNA-PKcs in kidney tissues of DNA-PKcs knockout and WT control mice (BALB/c background) subjected to UUO (7d).

9. Fig 2 h,I,j and k are not labeled correctly.

Response: Thank you very much for your careful review. We apologize that Fig 2 h,I,j

and k were not labeled correctly and we have corrected them in the revised manuscript.

10. In Fig 3, The western blot image for TGF β 1 stimulated DNA-PK expression and phosphorylation in cultured cells should be presented.

Response: The Western blot images for TGF β 1-stimulated DNA-PKcs expression and phosphorylation in HK2 cells (**Fig. 4i**) and NRK-49F cells (**Figure 4j**) are presented in the revised manuscript.

Reviewer #3 (Remarks to the Author):

The authors begin by showing that DNA-PK expression and the presence of its S2056 phosphorylated form is markedly increased in the kidneys of CKD patients and CKD-model mice. They go on to show that DNA-PK knockout or treatment with the DNA-PKcs inhibitor, NU7441, attenuates CKD in mice, and initiate in the next step in vitro experiments to uncover the underlying molecular mechanisms aiming at novel therapies for the disease.

In the in vitro part of their work the authors show that DNA-PK deficiency preserves epithelial cell phenotype and inhibits fibroblast activation caused by TGF β 1. This is relevant because in injured kidneys, interstitial fibrosis involves TGF β 1 induction, that causes epithelial cell dedifferentiation and myofibroblast activation. They present evidence that a putative DNA-PK substrate, TAF7, promotes mTORC1 activation by upregulating the expression of RAPTOR, which in-turn promotes metabolic reprogramming in injured epithelial cells and myofibroblasts. They demonstrate that inhibition of DNA-PK corrects metabolic reprogramming by regulating

TAF7/mTORC1 signaling in CKD, offering new ways for treating the disease.

The paper presents interesting and novel results that uncover unknown facets of DNA-PK function. The possibility that DNA-PK inhibitors may help to treat the disease is also exciting and worth pursuing. The paper summarizes a great deal of work that appears well conducted, and is overall very well written. The results shown justify the conclusions drawn, as the reported effects are uniformly large.

Specific comments:

1. From the perspective of this reviewer one missing aspect in the work concerns the mode of DNA-PK activation. As the authors point out, the typical activator, DSBs, appear not to be involved in the observed effects. It will certainly be unrealistic to ask the authors to delve in this direction as part of the present paper, however, one aspect that could certainly be added without much effort and which will significantly enhance the message of the paper, is to test whether KU is actually contributing to the observed effects. The authors, following the dogma of the field, assume that this is the case, i.e. that KU is required. However, because γ H2AX foci are not seen at any stage of the disease or in the extensive set of the reported in vitro experiments, it will be tantalizing to investigate whether DNA-PKcs actually functions in this setting independently of KU. This could be readily investigated in a set of knockdown experiments, where KU components of knocked down and selected endpoints investigated.

Response: Thank you very much for your valuable suggestion. According to your suggestion, we constructed KU80 knockout tubular epithelial cells by CRISPR/cas9 and the results showed KU80 knockout boosted FN production induced by TGF β 1 in

tubular epithelial cells, which was opposite to the outcome of DNA-PKcs knockout (**Fig. 4m**). KU70/80 firstly recognizes DNA broken ends at DSB, then recruits DNA-PKcs to form DNA-PK that is necessary in the repair of DSB via NHEJ⁵. These results suggest that DNA-PKcs may mediate profibrotic response of renal epithelial cells independent of its DSB repair function. For this reason, we revised DNA-PK as DNA-PKcs in the Title of manuscript.

2. The connection between DNA-PKcs and the activation of TGF β 1 could be analyzed in greater detail. DNA-PKcs has been shown to interact with internalized growth factors and it will be interesting to see whether this is also the case with TGF β 1.

Response: Thanks much for the valuable suggestion. To analyze the connection between DNA-PKcs and the activation of TGF β 1, we also examined the possible mechanism of TGF β 1-induced DNA-PKcs expression. Firstly, we found multiple possible binding sites of SMAD2 in the promoter region of DNA-PKcs through JASPAR transcription factor binding profiles analysis⁴. Secondly, luciferase assay results showed that overexpression of SMAD2 enhanced transcriptional activation of human DNA-PKcs (**Supplementary Fig. 5e**), suggesting a transcriptional regulation of SMAD2 on DNA-PKcs. Additionally, the results showed DNA-PKcs knockout inhibited the activation of SMAD2/SMAD3 in UUO mouse model (**Supplementary Fig. 5f**). All of these results suggest DNA-PKcs-SMAD signaling perhaps form a positive loop under CKD condition (**Supplementary Fig. 5g**).

3. The broad effects of systemic DNA-PKcs ablation in the mouse leave room for indirect contributions to the observed endpoints. The authors address this point in the

Discussion. Have the authors also considered organ specific ablation of the protein – for example in the kidney. Such a mouse model will provide a perfect tool for experiments designed to develop therapies for CKD. The authors may wish to discuss this aspect.

Response: To exclude the potential effect of lymphocyte deficiency, we constructed proximal renal tubular epithelial cells with specific DNA-PKcs knockout *in vivo* by using CRISPR/cas9 knockin mice (**Supplementary Fig. 2e, f**), as described in the Methods section. Three weeks after subcapsular injection of adeno-associated virus (AAV) containing guide RNA (gRNA) for DNA-PKcs, the proximal renal tubular specific DNA-PKcs knockout mice were subjected to UUO and euthanized at 7 days after surgery. The immunofluorescence staining of DNA-PKcs showed knockout of DNA-PKcs in renal tubular cells of cas9 mice with the injection of AAV-sgRNA subjected to UUO, compared to cas9 mice injected with an AAV without gRNA for DNA-PKcs (**Fig. 2h**). Protein levels of DNA-PKcs and profibrotic markers, including FN and α -SMA, were also significantly reduced in the UUO kidney tissues of cas9 mice injected with AAV-sgRNA (**Fig. 2g**).

4. Thinking about therapeutic application of the information in the paper: can the authors speculate where they hope to get more mileage: Inhibition of DNA-PK, inhibition of TGF β 1, or inhibition of mTor?

Response: Thank you very much for your kind comments. As mTOR plays a central role in maintaining metabolic homeostasis, inhibitors of mTOR sometimes show adverse side effects^{11,12}. Moreover, long-term inhibition of TGF- β signaling could also

be associated with unacceptable adverse side effects ¹³. In this study, we found that the activity of DNA-PKcs was almost undetectable in normal human kidney tissues, but greatly upregulated in human CKD kidney tissues. Inhibition of DNA-PKcs corrected metabolic reprogramming in fibrotic kidneys, providing a promising strategy for the treatment of CKD. All of these results suggest inhibition of DNA-PKcs could be more effective for renal fibrosis than inhibition of TGFβ1 or mTOR, although more research is still needed to validate this possibility. This part has been added into the Discussion section and highlighted in red.

Reviewer #4 (Remarks to the Author):

1. The manuscript by Jia et al. describes observations that DNA-PK is upstream of CKD via a TAF7/RAPTOR/mTOR signalling process. The authors claim that TAF7 is a phosphorylation substrate of DNA-PK. The evidence for this is not sufficiently investigated. First, the authors performed phosphoproteomics analysis in DNA-PK deficient mice model compared to control. The technical details for phosphopeptide and phosphorylation site identification are not provided. Typically, phosphoproteins and the identification of phosphorylation sites require phosphoprotein/peptide enrichment because of low-medium abundance.

Response: Thank you very much for your kind comments. We apologize that we did not provide the technical details for phosphopeptide and phosphorylation site identification. Now, the technical details for phosphoprotein/peptide enrichment and phosphorylation site identification were provided in **Phosphoproteomics analysis in**

the Methods section of the revised manuscript and highlighted in red.

2. It is not clear how and which phosphorylation sites in the kidney tissues were identified. It is claimed that among these proteins TAF7 phosphorylation was decreased. One can only guess that the authors refer to phosphorylation site 213, from figure 5. Furthermore, any up-and downregulation of TAF7 is then claimed by western blot analysis of the TAF7 protein and not the phosphorylated form by saying that the phosphorylated TAF7 is 55kDa instead of 40kDa referring to literature. However, from the cited literature this is not clear at all. A direct analysis of the phosphorylation site of TAF7 is necessary. The authors need to identify and quantify the direct phosphorylation site by mass spectrometry. It is also not validated by the data that DNA-PK phosphorylates TAF7, there is only indirect evidence. Here more *in vivo* and *in vitro* studies are necessary.

Response: Thank you very much for your kind comments. Firstly, Protein-Protein docking between DNA-PKcs and TAF7 was performed through an online database ClusPro^{14,15}. The docking results showed a possible direct interaction between DNA-PKcs and TAF7, and the 213-site serine of TAF7 is one of the nearest sites perhaps binding to DNA-PKcs (**Supplementary Fig. 6a**). An *in vitro* kinase assay was performed to analyze whether TAF7 was phosphorylated by DNA-PKcs directly. The products of kinase assay were analyzed by mass spectrometry. The results showed S213, T137, Y24 and S171 sites of human TAF7 were phosphorylated by DNA-PKcs (**Supplementary Fig. 6b**). Additionally, the phosphorylation of TAF7 in UUO mice was also confirmed by Phos-tag SDS-PAGE (**Supplementary Fig. 6c**). Phos-tag SDS-

PAGE was performed as previous reported^{16,17}. 100 μ M Phosbind (Apexbio) was added to 8% (w/v) polyacrylamide separating gels before polymerization, according to the manufacturer's instruction, to separate phosphorylated isoforms of TAF7. All of these results were provided in our revised manuscript and highlighted in red.

References

- 1 Zheng, B. *et al.* Over-expression of DNA-PKcs in renal cell carcinoma regulates mTORC2 activation, HIF-2alpha expression and cell proliferation. *Sci Rep* **6**, 29415, doi:10.1038/srep29415 (2016).
- 2 Blunt, T. *et al.* Identification of a nonsense mutation in the carboxyl-terminal region of DNA-dependent protein kinase catalytic subunit in the scid mouse. *Proceedings of the National Academy of Sciences of the United States of America* **93**, 10285-10290, doi:10.1073/pnas.93.19.10285 (1996).
- 3 Kurimasa, A. *et al.* Catalytic subunit of DNA-dependent protein kinase: impact on lymphocyte development and tumorigenesis. *Proc Natl Acad Sci U S A* **96**, 1403-1408, doi:10.1073/pnas.96.4.1403 (1999).
- 4 Castro-Mondragon, J. A. *et al.* JASPAR 2022: the 9th release of the open-access database of transcription factor binding profiles. *Nucleic Acids Res* **50**, D165-D173, doi:10.1093/nar/gkab1113 (2022).
- 5 Chaplin, A. K. *et al.* Dimers of DNA-PK create a stage for DNA double-strand break repair. *Nat Struct Mol Biol* **28**, 13-19, doi:10.1038/s41594-020-00517-x (2021).
- 6 Woodard, L. E. *et al.* Hydrodynamic Renal Pelvis Injection for Non-viral Expression of Proteins in the Kidney. *J Vis Exp*, doi:10.3791/56324 (2018).
- 7 Serra, N., Velte, E. K., Niedenberger, B. A., Kirsanov, O. & Geyer, C. B. The mTORC1 component RPTOR is required for maintenance of the foundational spermatogonial stem cell pool in micedagger. *Biol Reprod* **100**, 429-439, doi:10.1093/biolre/i0y198 (2019).
- 8 Ramirez-Valle, F., Badura, M. L., Braunstein, S., Narasimhan, M. & Schneider, R. J. Mitotic raptor promotes mTORC1 activity, G(2)/M cell cycle progression, and internal ribosome entry site-mediated mRNA translation. *Mol Cell Biol* **30**, 3151-3164, doi:10.1128/MCB.00322-09 (2010).
- 9 Yang, K. *et al.* T cell exit from quiescence and differentiation into Th2 cells depend on Raptor-mTORC1-mediated metabolic reprogramming. *Immunity* **39**, 1043-1056, doi:10.1016/j.immuni.2013.09.015 (2013).
- 10 Valencia, T. *et al.* Metabolic reprogramming of stromal fibroblasts through p62-mTORC1 signaling promotes inflammation and tumorigenesis. *Cancer Cell* **26**, 121-135, doi:10.1016/j.ccr.2014.05.004 (2014).

- 11 Nguyen, V. N., Abagyan, R. & Tsunoda, S. M. Mtor inhibitors associated with higher cardiovascular adverse events-A large population database analysis. *Clin Transplant* **35**, e14228, doi:10.1111/ctr.14228 (2021).
- 12 Pallet, N. & Legendre, C. Adverse events associated with mTOR inhibitors. *Expert Opin Drug Saf* **12**, 177-186, doi:10.1517/14740338.2013.752814 (2013).
- 13 Gyorfı, A. H., Matei, A. E. & Distler, J. H. W. Targeting TGF-beta signaling for the treatment of fibrosis. *Matrix Biol* **68-69**, 8-27, doi:10.1016/j.matbio.2017.12.016 (2018).
- 14 Desta, I. T., Porter, K. A., Xia, B., Kozakov, D. & Vajda, S. Performance and Its Limits in Rigid Body Protein-Protein Docking. *Structure* **28**, 1071-1081 e1073, doi:10.1016/j.str.2020.06.006 (2020).
- 15 Vajda, S. *et al.* New additions to the ClusPro server motivated by CAPRI. *Proteins* **85**, 435-444, doi:10.1002/prot.25219 (2017).
- 16 Qiu, W. & Steinberg, S. F. Phos-tag SDS-PAGE resolves agonist- and isoform-specific activation patterns for PKD2 and PKD3 in cardiomyocytes and cardiac fibroblasts. *J Mol Cell Cardiol* **99**, 14-22, doi:10.1016/j.yjmcc.2016.08.005 (2016).
- 17 Magliozzi, J. O. & Moseley, J. B. Pak1 kinase controls cell shape through ribonucleoprotein granules. *Elife* **10**, doi:10.7554/eLife.67648 (2021).

REVIEWER COMMENTS

Reviewer #1 (Remarks to the Author):

The authors have addressed all my previous concerns to satisfaction. I have no further comments.

Reviewer #2 (Remarks to the Author):

Although the authors have addressed some concerns, some critical issues are still needed to be clarified.

1. DNA-PKcs was induced in both tubular cells as well as interstitial cells after UUO or IRI, while in the tubular cell specific deletion of DNA-PKcs kidneys, it looks DNA-PKcs was disappeared in the whole kidney tissue (Figure 2H), but not tubular cell specific deletion. The reviewer suggests that more evidences should be provided to show the specific deletion of DNA-PKcs in their genetic model.
2. Wang S, et al has reported that DNA-PKcs interacts with and phosphorylates Fis1 to induce mitochondrial fragmentation in tubular cells during acute kidney injury. This paper is not cited. (Sci Signal. 2022 Mar 15;15(725):eabh1121. doi: 10.1126/scisignal.abh1121. Epub 2022 Mar 15.) In addition, since mitochondrial fragmentation plays a crucial role in the kidney fibrogenesis, the role for Fis1 which mediates DNA-PKcs action during AKI in their fibrosis model should be tested.
3. The author claimed that human DNA-PKcs overexpression plasmids (Addgene: 83317) were delivered into mouse kidneys through a tail vein high-pressure injection method. Although tail vein high-pressure injection of naked plasmid has been reported to induce overexpression of exogenous gene in the liver of mouse model, it is highly impossible to induce exogenous gene expression in the kidneys. It is of note that the reference cited by the authors "Hydrodynamic Renal Pelvis Injection for Non-viral Expression of Proteins in the Kidney. J Vis Exp, doi:10.3791/56324 (2018) ", which is through pelvis injection, not tail vein used by the author in their work. The authors should clarify this.

Reviewer #3 (Remarks to the Author):

The authors have done a wonderful job in revising the manuscript.

Reviewer #4 (Remarks to the Author):

The authors have addressed the concerns. The manuscript has improved by more detailed description of the methods.

Response to reviewers:

Reviewer #1 (Remarks to the Author):

The authors have addressed all my previous concerns to satisfaction. I have no further comments.

Response: Thank you very much for your valuable comments and support on this research work.

Reviewer #2 (Remarks to the Author):

Although the authors have addressed some concerns, some critical issues are still needed to be clarified.

1. DNA-PKcs was induced in both tubular cells as well as interstitial cells after UUO or IRI, while in the tubular cell specific deletion of DNA-PKcs kidneys, it looks DNA-PKcs was disappeared in the whole kidney tissue (Figure 2H), but not tubular cell specific deletion. The reviewer suggests that more evidences should be provided to show the specific deletion of DNA-PKcs in their genetic model.

Response: Thank you very much for your valuable comments. In order to confirm the specific deletion of DNA-PKcs in renal tubular cells of DNA-PKcs^{TEC KO} mice, immunofluorescence co-staining of DNA-PKcs with α -SMA (a marker of myofibroblasts) was performed. The results showed DNA-PKcs was still expression in α -SMA positive myofibroblasts (see below figure) and these results were also added in the revised manuscript (**Supplementary Fig. 2g**). It should be noted that the proliferation and activation of myofibroblasts in CKD kidneys were decreased when

renal tubular injury was improved as described in previous study¹. Thus, the lower number of α -SMA-positive cells in UUO kidneys of DNA-PKcs TEC KO mice should be resulted from the attenuation of renal fibrosis.

Immunofluorescence staining of DNA-PKcs and α -SMA in kidneys of DNA-PKcs TEC KO and control mice subjected to UUO (day 7), scale bar: 20 μ m.

2. Wang S, et al has reported that DNA-PKcs interacts with and phosphorylates Fis1 to induce mitochondrial fragmentation in tubular cells during acute kidney injury. This paper is not cited. (Sci Signal. 2022 Mar 15;15(725):eabh1121. doi: 10.1126/scisignal.abh1121. Epub 2022 Mar 15.) In addition, since mitochondrial fragmentation plays a crucial role in the kidney fibrogenesis, the role for Fis1 which mediates DNA-PKcs action during AKI in their fibrosis model should be tested.

Response: The study of Wang S, et al was published after the submission of our manuscript, thus, we didn't renew our references in time. Now, this paper was cited in

our revised manuscript (ref. 43). Wang S, et al found cytoplasmic DNA-PKcs was increased in mouse kidney tissues with AKI. Increased cytoplasmic DNA-PKcs interacted with Fis1 and phosphorylated it at Thr34 in its TQ motif, which increased the affinity of Fis1 for Drp1 and induced mitochondrial fragmentation. Following your important comment, we investigated whether Fis1 mediated profibrotic effects of DNA-PKcs in kidney fibrosis model. First, nuclear extraction protein analysis was performed and the results showed DNA-PKcs was mainly localized and increased in the nucleus in kidneys of UUO mice which was different from AKI model (below figure a, **Supplementary Fig. 6k**). Second, we knocked down Fis1 in renal tubular epithelial cells *in vitro*, and the cells were treated with TGF β 1. Western blot analysis showed Fis1 protein levels were not induced by TGF β 1 in tubular epithelial cells, and knock down of Fis1 exacerbated profibrotic response of tubular epithelial cells challenged by TGF β 1 (below figure b, **Supplementary Fig. 6l**). These results related to Fis1 did not support a contribution of Fis1 in mediating the profibrotic role of DNA-PKcs in CKD, indicating a different role of Fis1 in different diseases and pathogenesis.

(a) The expression and subcellular location of DNA-PKcs were measured in kidneys after UUO for 7 days through fractionated western blotting. (b) Protein levels of FN and FIS1 were analyzed by western blot in tubular epithelial cells transfected with si-NC or si-FIS1 then treated with TGF β 1 (5 ng/ml) for 24 h.

3. The author claimed that human DNA-PKcs overexpression plasmids (Addgene: 83317) were delivered into mouse kidneys through a tail vein high-pressure injection method. Although tail vein high-pressure injection of naked plasmid has been reported to induce overexpression of exogenous gene in the liver of mouse model, it is highly impossible to induce exogenous gene expression in the kidneys. It is of note that the reference cited by the authors “Hydrodynamic Renal Pelvis Injection for Non-viral Expression of Proteins in the Kidney. *J Vis Exp*, doi:10.3791/56324 (2018)”, which is through pelvis injection, not tail vein used by the author in their work. The authors should clarify this.

Response: Thank you very much for your valuable comments. As pointed out above, hydrodynamic tail vein injection of naked plasmids primarily transfected exogenous genes in liver. At the same time, this method also can deliver genes to renal tubular cells as confirmed by our and other groups²⁻⁶. In this study, our results showed that in kidney, exogenous DNA-PKcs can be expressed in renal tubular cells via this tail vein high-pressure injection method. However, we cited a wrong reference during the preparation of the manuscript. We appreciate your careful review avoiding a mistake in reference citation. We apologize for this mistake and the correct reference was included in the revised manuscript.

Reviewer #3 (Remarks to the Author):

The authors have done a wonderful job in revising the manuscript.

Response: Thank you very much for your valuable comments and support on this research work.

Reviewer #4 (Remarks to the Author):

The authors have addressed the concerns. The manuscript has improved by more detailed description of the methods.

Response: Thank you very much for your valuable comments and support on this research work.

References

- 1 Li, H. *et al.* Upregulation of HER2 in tubular epithelial cell drives fibroblast activation and renal fibrosis. *Kidney international* **96**, 674-688, doi:10.1016/j.kint.2019.04.012 (2019).
- 2 Yu, X. *et al.* Nuclear receptor PXR targets AKR1B7 to protect mitochondrial metabolism and renal function in AKI. *Science translational medicine* **12**, doi:10.1126/scitranslmed.aay7591 (2020).
- 3 Peng, W. *et al.* BMP-7 ameliorates partial epithelial-mesenchymal transition by restoring SnoN protein level via Smad1/5 pathway in diabetic kidney disease. *Cell death & disease* **13**, 254, doi:10.1038/s41419-022-04529-x (2022).
- 4 Mao, Y. *et al.* EI24 alleviates renal interstitial fibrosis through inhibition of epithelial-mesenchymal transition and fibroblast activation. *FASEB journal : official publication of the Federation of American Societies for Experimental Biology* **35**, e21239, doi:10.1096/fj.202002089R (2021).
- 5 Li, Y. *et al.* Activation of GSDMD contributes to acute kidney injury induced by cisplatin. *American journal of physiology. Renal physiology* **318**, F96-F106, doi:10.1152/ajprenal.00351.2019 (2020).
- 6 Romero-Vasquez, F. *et al.* Overexpression of HGF transgene attenuates renal inflammatory mediators, Na(+)-ATPase activity and hypertension in spontaneously hypertensive rats. *Biochimica et biophysica acta* **1822**, 1590-1599, doi:10.1016/j.bbadis.2012.06.006 (2012).

REVIEWERS' COMMENTS

Reviewer #2 (Remarks to the Author):

The authors have addressed the concerns.

Response to reviewers:

Reviewer #2 (Remarks to the Author):

The authors have addressed the concerns.

Response: Thank you very much for your valuable comments and support on this research work.